# Evolution of giant pandoravirus revealed by CRISPR/Cas9

Hugo Bisio[1,2] ✉, Matthieu Legendre [1,2], Claire Giry [1], Nadege Philippe[1], Jean-Marie Alempic[1], Sandra Jeudy [1] & Chantal Abergel [1] ✉

Giant viruses (GVs) are a hotspot of unresolved controversies since their discovery, including the definition of "Virus" and their origin. While increasing knowledge of genome diversity has accumulated, GV functional genomics was largely neglected. Here, we describe an experimental framework to genetically modify nuclear GVs and their host *Acanthamoeba castellanii* using CRISPR/Cas9, shedding light on the evolution from small icosahedral viruses to amphora-shaped GVs. Ablation of the icosahedral major capsid protein in the phylogenetically-related mollivirus highlights a transition in virion shape and size. We additionally demonstrate the existence of a reduced core essential genome in pandoravirus, reminiscent of their proposed smaller ancestors. This proposed genetic expansion led to increased genome robustness, indicating selective pressures for adaptation to uncertain environments. Overall, we introduce new tools for manipulation of the unexplored genome of nuclear GVs and provide experimental evidence suggesting that viral gigantism has aroused as an emerging trait.

Giant viruses (GV) have genomes up to 2.8 megabases and form viral particles that can match in size some cellular organisms[1,2]. Their dsDNA genomes can encode more than 1000 genes, 70% of them corresponding to proteins unseen in any other organisms[2,3]. These genes are referred to as ORFans[4]. In addition, these atypical viruses encode proteins involved in translation and energy metabolism, hallmark functions of the cellular world[5–7]. These unprecedented features have led to great controversies regarding the origin of such viruses[8]. The origin of these viruses has been attributed either to reductive evolution branching from an ancestral cell[8] or evolution towards complexity from smaller viruses[9].

Particularly, pandoraviruses possess the current record in genome size of the viral world with P. salinus harboring a double-stranded DNA molecule of 2.77 Mbp[1]. The particles of these viruses are amphora-shaped and formed by a membranous compartment encapsulated by a tegument-like envelope made of several layers[1,10,11], one of them composed of cellulose[11]. An apical pore allows delivery of the content of the particle by fusion of the internal membrane with the phagosome[1]. Upon delivery of the virion content, pandoravirus enters an eclipse phase of 2 to 4hs, followed by a dramatic host nucleus reorganization (including shape, content and loss of membrane integrity)[1]. Eight to ten hours post-infection (PI), host cells round up and new virions become visible in the infected cell[1]. Interestingly, the tegument of pandoravirus particles and the content of the virion seem to be synthesized/assembled simultaneously, this process being initiated at the apical pore[1]. Virions are released by cell lysis or exocytosis[1,10].

Here, we developed a battery of genetic tools to study nuclear giant viruses and their host *Acanthamoeba castellanii*. In addition, we used this technology to tentatively trace back the evolution from a small icosahedral to the amphora-shaped pandoravirus GV[12]. We demonstrate the existence of an essential core region in pandoravirus, which corresponds with the accumulation of evolutionary conserved genes across their proposed smaller viral ancestors. Finally, we demonstrate the essentiality of the icosahedral Major Capsid Protein in mollivirus, a phylogenetically related virus and previously proposed

[1]Aix–Marseille University, Centre National de la Recherche Scientifique, Information Genomique & Structurale, Unite Mixte de Recherche 7256 (Institut de Microbiologie de la Mediterranee, FR3479, IM2B), 13288, Marseille Cedex 9, France. [2]These authors contributed equally: Hugo Bisio, Matthieu Legendre. ✉e-mail: hugo.bisio@igs.cnrs-mrs.fr; chantal.abergel@igs.cnrs-mrs.fr

evolutionary intermediate in virion shape[9,13,14]. Overall, our analysis demonstrates the potential of genetic screens to uncover conserved biological processes and host–pathogen interactions for the most genetically complex viruses discovered so far.

## Results

### CRISPR/Cas9 expression allows efficient modification of host DNA

To achieve genetic manipulation of GVs, we built an *Acanthamoeba castellani* CRISPR/Cas9 system, using a plasmid encoding a polycistronic tRNA-gRNA[15,16] and the *Sp*Cas9 gene fused to the green fluorescent protein (GFP)[17] gene (Fig. 1a). To test the system, we conducted Cas9-mediated modification of an episomally encoded red fluorescent protein (mRFP) using the vector vc241 (Fig. S1a). When amoebas were transfected with Cas9 and specific guides targeting the mRFP gene (Fig. S1a), we observed a significant decrease in the mRFP positive population, which correlated with Cas9 expression (GFP

positive) (Fig. 1b, c). The presence of double-negative amoebas is likely explained by the amitotic nuclear division of these organisms, leading to aneuploidy and the potential loss of genetic material[18,19]. Double-labeled amoebas were not observed when on-target guides were present (Fig. 1b–c). The locus targeted by the gRNA was amplified by PCR, cloned into a TA vector and single clones were isolated for sanger sequencing. Accordingly, genotyping confirmed gene locus modification in ~50% of the screened library (10 clones) (Fig. 1d and S1b). Importantly, the expression of Cas9 in presence or absence of off-target gRNA did not affect the doubling time of the amoeba (Fig S1c) or increased mortality rate (Fig S1d). To increase the efficiency of genome modification, we hypothesized that the introduction of a "mutagenic chain reaction" (Fig. S1e)[20] would allow the propagation of recombinant genes. Concordantly, 100% reduction of mRFP expression was demonstrated when this strategy was utilized (Fig. 1b–c and S1f). Correct integration and homozygosis were validated by PCR (Fig. 1f). No recombinant cells were detected when off-target guides were

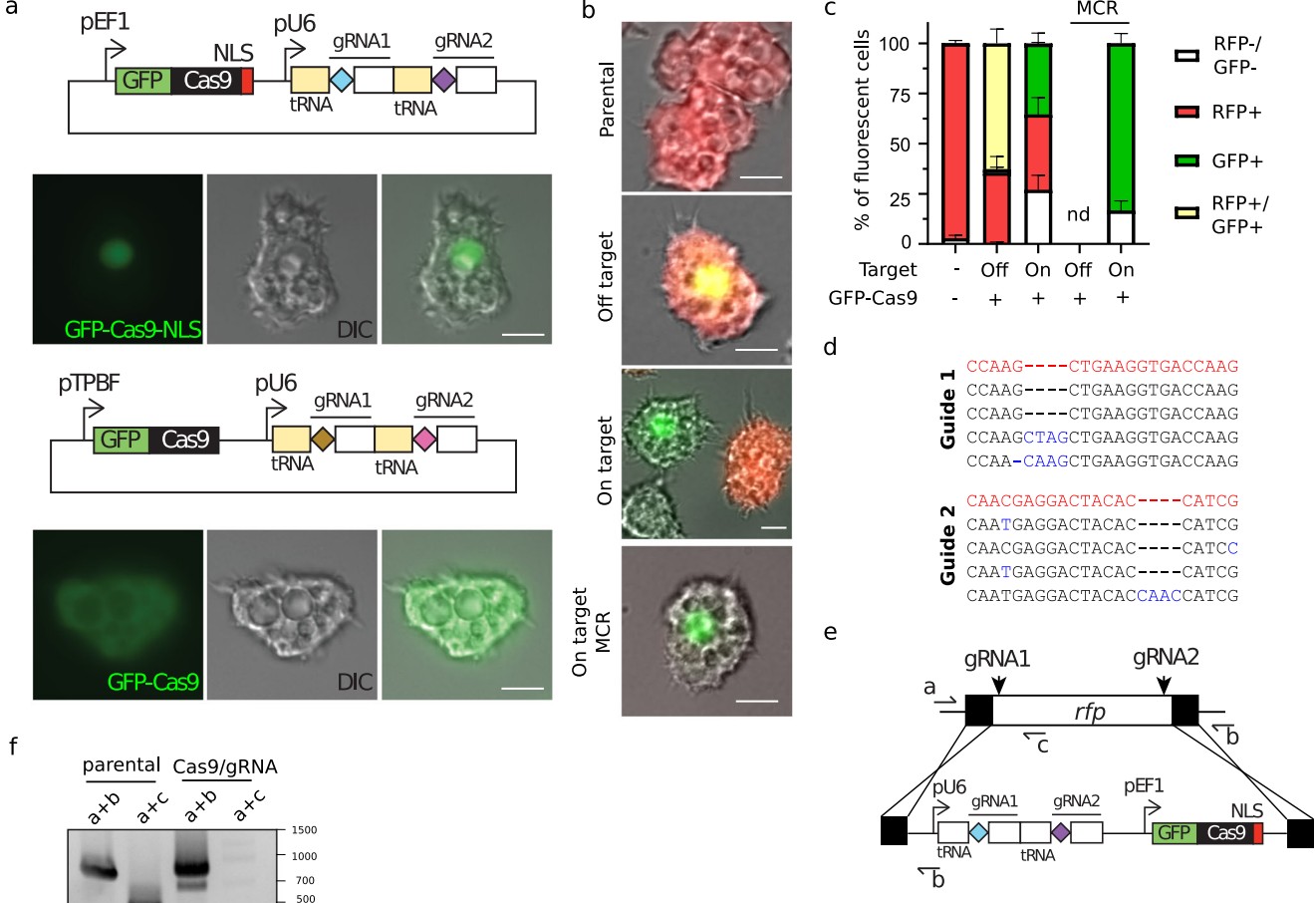

**Fig. 1 | CRISPR/Cas9 allows manipulation of *A. castellanii*. a** Constructs used to constitutively express Cas9 in *A. castellanii*. The sequence of the gRNA is depicted with diamonds followed by the Cas9-binding scaffold and an *A. castellanii* tRNA (yellow rectangles). The localization of the GFP-Cas9 fusion for the different constructs is shown by fluorescence. Micrographs are representative of 3 independent experiments. Scale bar: 10μm. **b** Representative micrographs showing amoebas expressing mRFP (vector Vc241) and GFP-Cas9 after selection for 2–3 weeks with the appropriate drug(s) (refer to materials and methods). mRFP (product of the targeted gene) and GFP (Indicating Cas9 expression) fluorescence are shown. MCR: mutagenic chain reaction. Scale bar: 10μm. **c** The quantification of the micrograph shown in (**b**). The mean ± SD of at least 200 amoebas (3 independent experiments ($n = 3$)) is shown. Amoebas were classified either as non-fluorescent, mRFP+, GFP+ or GFP+mRFP+. MCR: mutagenic chain reaction. Guides targeting pandoravirus

rpb1 were used as off-target gRNAs. Nd: not detected. **d** Representative sequencing results of targeted guide sequences on *rfp* upon transfection with on-target gRNAs. PCR were performed on the target sequences as shown in Fig. S1a, b, cloned into a TA cloning vector and single clones were sent for sequencing. The wild type sequence and mutations generated are shown in red and blue respectively. Ten individual clones were amplified and sequenced. **e** Schematic representation of the rfp locus, guide targeting location, homology arms for recombination and primer annealing sites for a disruption using a "mutagenic chain reaction" strategy (see Fig. S1e). **f** Gene disruption of *rfp* by the "mutagenic chain reaction" observed at the population level after 2–3 weeks post-transfection. Expected PCR size: a + b: 837 bp (unmodified locus), a + b 890 bp (recombinant locus), a + c: 500 bp (unmodified locus). Note that the primer b anneals both in the wild type and recombinant locus resulting in PCR products with slight differences in size.

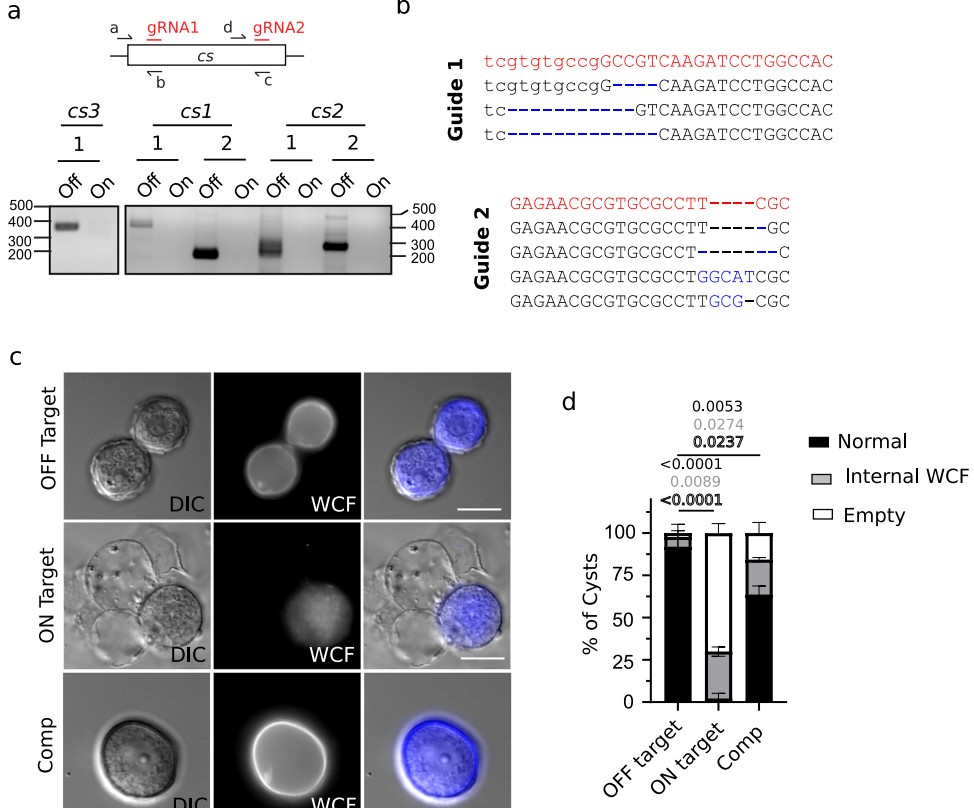

**Fig. 2 | CRISPR/Cas9 allows manipulation of *A. castellanii*. a** Disruption of genes encoding cellulose synthase 1–3 (*cs1-3*) is observed at the population level after 2–3 weeks post-transfection. Reverse primers are designed to anneal at the gRNA targeting sequencing. Disappearance of the PCR product correlates with modification of the locus in all the alleles. Expected PCR size: *cs1*: 430 bp (guide 1), 200 bp (guide 2). *cs2*: 250 bp (guide 1) 265 bp (guide 2). *cs3*: 320 bp. Guides targeting mRFP were used as off-target gRNAs. **b** Representative sequencing results of *cs1* upon transfection with on-target gRNAs. PCR were performed on the target sequences, cloned into a TA cloning vector and single clones were sent for sequencing. The wild-type sequence and mutations generated are shown in red and blue, respectively. Ten individual clones were amplified and sequenced. **c** Depletion of the cellulose synthase impedes cellulose deposition at the periphery of *A. castellanii* cyst. WCF: Calcofluor white, which stains cellulose. Scale bar: 20μm. **d** The quantification of the micrograph shown in (**c**). The mean ± SD of at least 200 amoebas (3 independent transfections (*n* = 3)) is shown. Amoebas were classified either as empty, normal or abnormal internal WCF staining. The null hypothesis (α = 0.05) was tested using unpaired two-tailed Student's *t* tests. Source data are provided as a Source Data file.

transfected (Fig. 1b, c), indicating that homologous recombination (HR) is rather inefficient in absence of double strand breaks.

Importantly, the genome of *A. castellanii* is highly polyploid (aprox. 25n[21], (Fig. S1e)), which could make genetic manipulation difficult. To assess the efficiency of Cas9 to modify the genome of *A. castellanii*, we designed gRNA targeting the cellulose synthase genes. *A. castellanii* encodes three possible genes for cellulose synthase (CS)[22], two of them containing a full sequence of the CS monomer. Concordantly, we designed two gRNA targeting conserved regions in the two full-length *CS* (*cs1* and *cs2*) and one in the shorter one (*cs3*) (Fig. 2a). Expression of Cas9 led to complete modification of the targeting sequencing as demonstrated by PCR using reverse primers annealing on the gRNA targeting sequences (Fig. 2a). These results indicate that each paralogue gene, present in approximately 25 copies[21], was fully targeted by Cas9 and modified by NHEJ (around 125 modified sites). Moreover, the locus targeted by the gRNA was amplified by PCR, cloned into a TA vector and 10 single clones were isolated for Sanger sequencing. Genotyping of this screened library confirmed gene locus modification in all clones of *cs1* (Fig. 2b). Importantly, due to the heterozygosis shown by the library screening and, since not every modification resulted in a frame shift for translation, full depletion of gene expression cannot be confirmed. While no changes in the doubling time of trophozoites were observed (Fig. S1f), a strong inhibition of encystment was induced upon knock-out/downregulation of CS, as demonstrated by SDS resistance (Fig. S1g)

and calcofluor (CFW) staining of cysts (Fig. 2c, d). These data reproduced previous results obtained by RNA silencing[23,24]. This phenotype was rescued by the expression of a Cas9-targeted resistant *cs1* gene fused with GFP (Fig. 2c, d and Fig S1h). Moreover, similar defects were observed at the ultrastructure level when analyzed by electron microscopy (Fig. S1j). These differences included the presence of opened pre-cysts with no cellular content, or cysts lacking a cyst endolayer, which is majorly composed of cellulose[23,24].

Thus, while these experiments demonstrate the potential for the use of CRISPR/Cas9 to modify the highly complex genome of *A. castellanii*, future efforts must be directed into optimizing vectors and protocols to allow controllable modifications and/or homozygosis in the cells.

## CRISPR/Cas9 expression allows efficient modification of viral DNA

Most giant viruses encode an RNA polymerase (RNAP), which modeled the RNAPII of modern eukaryotes[25]. Due to its likely essentiality for viral replication, we targeted the *rpb1* gene to assess the capacity of Cas9 to target viral genomes (Fig. S2a). Nuclear Cas9 expression in amoeba cells (Fig. 3a) showed efficient regulation of *Pandoravirus neocaledonia*[3] and *Mollivirus kamchatka*[26] replication (Fig. 3a, b), while the cytosolic version of the protein (Fig. 3a) was unable to target the genome of *Pithovirus sibericum*[27] or *Mimivirus reunion*[28] (Fig. 3a, b). The incapability of Cas9 to target the genome

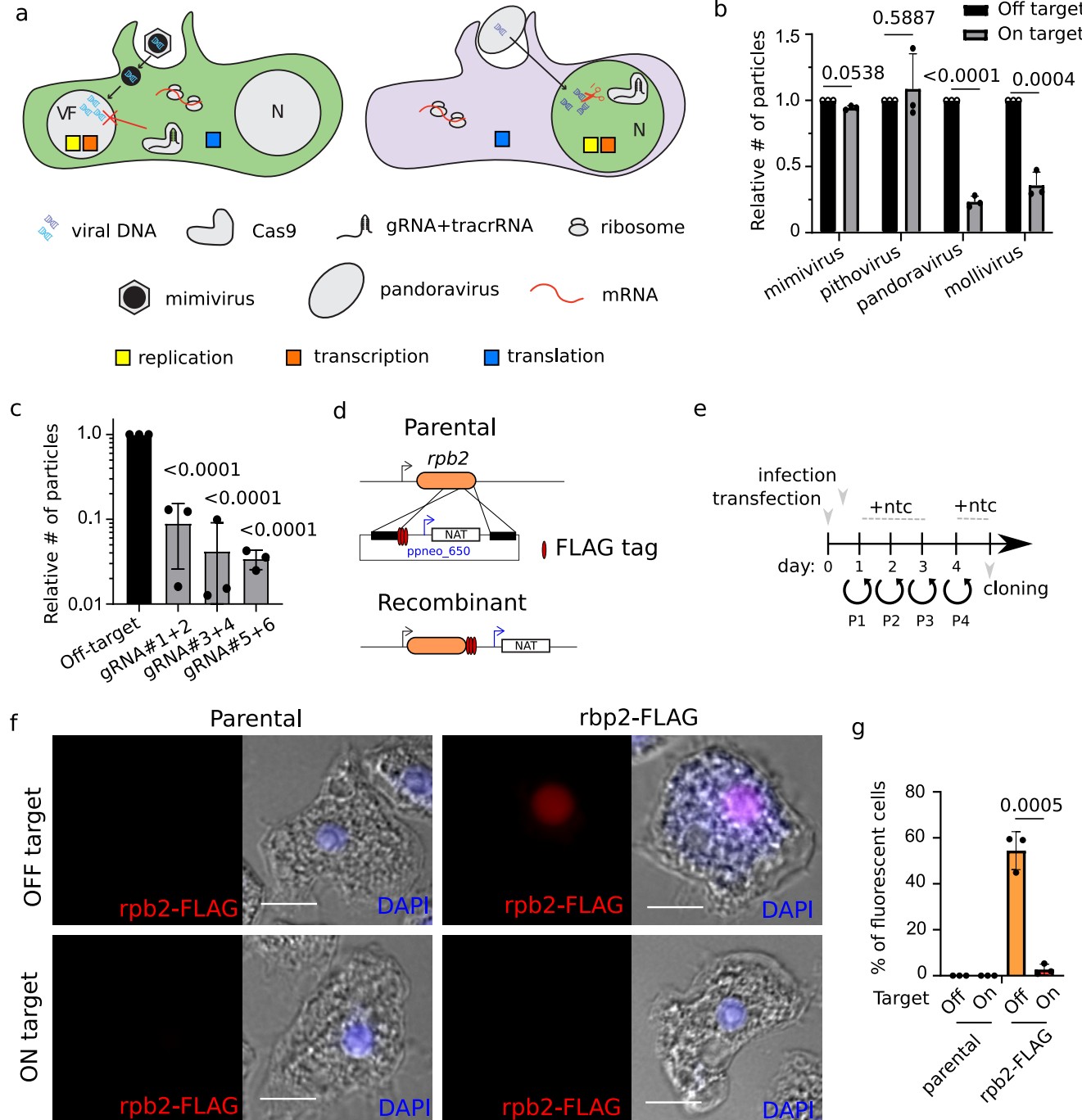

**Fig. 3 | CRISPR/Cas9 allows manipulation of nuclear GVs. a** Schematic depiction of genetic manipulation of GVs. *A. castellanii* expressing a nuclear or cytoplasmic Cas9 are shown in the left and right, respectively. VF: viral factory. N: Nucleus. **b** Quantification of the number (#) of viruses produced 24 h post-infection (hpi) of CRISPR/Cas9 expressing amoebas. Values are expressed relative to # of viral particles produced within amoebas expressing off-target gRNAs. A cytoplasmic version of Cas9 was used to attempt gene disruption of cytoplasmic GV (*Mimivirus reunion*[28] and *Pithovirus sibericum*[27]) and a nuclear version of Cas9 was used for the gene disruption of nuclear GVs (*Pandoravirus neocaledonia*[3] and *Mollivirus kamchatka*[26]). Data correspond to the mean ± SD of 3 independent experiments. MOI = 1. Guides targeting mimivirus rpb1 were used as off-target gRNAs for pithovirus infection and vice versa. Guides targeting pandoravirus rpb1 were used as off-target gRNAs for the infection of mollivirus and vice versa. The null hypothesis (α = 0.05) was tested using unpaired two-tailed Student's *t* tests. **c** Multiple gRNA combinations were used to target the *rpb1* gene of *Pandoravirus neocaledonia*. Data

were performed as Fig. 2b. Guides targeting mollivirus *mcp* were used as off-target gRNAs. Sequence of the gRNA 1 to 6 are shown in Table S1. The null hypothesis (α = 0.05) was tested using unpaired two-tailed Student's *t* tests. **d** Schematic representation of the vector and strategy utilized for rbp2 endogenous tagging in *Pandoravirus neocaledonia*. Selection cassette was introduced by homologous recombination. **e** Cartoon depicting the strategy for selection of recombinant viruses. Viral infection was performed 1 h post-transfection. Ntc: Nourseothricin. P = passage. **f** Nuclear localization of endogenously tagged RPB2-FLAG shown by immunofluorescence assay using anti-FLAG antibodies. DAPI: nuclear marker. The immunofluorescence signal disappears upon infection of cells encoding CRISPR/Cas9 targeting the *rpb2* locus. Guides targeting mollivirus *mcp* were used as off-target gRNAs. Scale bar: 10 μm. **g** The quantification of the micrograph shown in (**f**). The mean ± SD of at least 200 amoebas (3 independent experiments (*n* = 3)) is shown. The null hypothesis (α = 0.05) was tested using unpaired two-tailed Student's *t* tests. Source data are provided as a Source Data file.

of strictly cytoplasmic viruses might reside in the shielding of the DNA by these viruses, excluding GFP-Cas9 from the viral factory as shown by fluorescence (Fig. S2b). A similar strategy was previously demonstrated for some bacteriophages[29]. Coherently with the likely essentiality of the *rnap*, only virions containing wild-type genomes (which escaped Cas9 targeting) were recovered in genomic libraries (Fig. S2c).

The efficient targeting of *rpb1* of pandoravirus was further confirmed by utilizing multiple gRNA, which rendered similar phenotypic consequences (Fig. 3c). Moreover, to confirm the depletion of the target proteins upon knock-out, we built a plasmid that allows endogenous tagging of targeted genes (Fig. 3d). This plasmid contains a triple HA tag or triple FLAG tag and a nourseothricin N-acetyl transferase (NAT) resistance cassette, which confers resistance to nourseothricin[30]. The NAT selection cassette was expressed under the control of a pandoravirus promoter (Fig. 3d). NAT selection was used to enrich the recombinant viruses, which were later cloned to obtain homogeneous populations (Fig. 3e and Fig. S2d). We opted to tag the rpb2 subunit since a similar strategy was previously used in poxviruses without phenotypic consequences[31]. As a result, while pandoravirus rpb2-3HA was easily visualized by immunofluorescence in cells infected with recombinant viruses, targeting of the gene by CRISPR/Cas9 led to the disappearance of the protein indicating successful depletion (Fig. 3f, g).

## A CRISPR/Cas9 genetic screen reveals a core essential genome in pandoravirus

To gather information on the repair mechanisms of double-strand breaks in nuclear GVs, we targeted genes with suspected different degrees of importance for virus replication. Fitness reduction of pandoravirus was highly dependent on the targeted gene locus (Fig. 4a) and ranged from 100-times reduction in particle numbers when genes at the 5' of the genome were targeted, to no fitness cost when genes at the 3' of the genome were targeted. Surprisingly, targeting of genes at the 3' of the genome resulted in large deletions rather than discrete modification of the targeted locus (Fig. 4b–d and S3a), suggesting these viruses are not efficient in repairing double-strand breaks. Quantification of the ratio between PCR products using 5' and 3' regions of the genome as templates demonstrates that large deletions are rather frequent at the 3' of the genome, but not detectable at the 5' (Fig. 4b). Moreover, viruses harboring such 3' large deletions were able to complete the full cycle of replication since cloning was achievable (Fig. 4c). Unexpectedly, pandoravirus was capable of losing at least over 600,000 bp (Δ1195458–1838505 base pairs) and more than 300 genes (Δ*pneo_612-974*) (Supplementary Data 1), as shown by pulsed-field gel electrophoresis (Fig. 4d) and PCR amplification of specific loci (Fig. S3a). Interestingly, while the deletion of almost one-third of the genome led to a significant reduction of viral fitness (Fig. 4e), no changes in virion morphology were observed (Fig. 4f). These data strongly correlate with the strikingly low abundance of virion proteins encoded by genes located at the 3' end of the genome (Fig. 4g). Moreover, viral DNA replication was significantly impaired in viruses lacking the genes present at the 3' of the genome (Fig. 4h). Taken together, these data demonstrate that these genes are important during the intracellular stage of the infectious cycle, most likely for host–pathogen interaction.

Interestingly, fitness differences during the first generation of viral production upon targeting with CRISPR/Cas9 were associated with the gene location rather than the gene target itself (Fig. 5a). These data indicate that large deletions were not viable when generated at the 5' end of the genome (core genome). The core genome includes the genes *pneo_77* (dispensable in blue, Fig. 5a) to *pneo_544* (essential in red, Fig. 5a). The identification of the precise limits of this area would require further characterization. Meanwhile, amplifications of the viruses produced under a constant selection with Cas9 targeting the

core genome, led occasionally to the production of resistant viruses (Fig. S3b), which contain modification of the genome at the target site (Fig. 5b). Coherently, shorter deletions were observed in viral progenies when genes present at the core genome were targeted (Fig. 5c), demonstrating the capability of these viruses to repair by non-homologous end joining, even if not efficiently. Taken together, we were able to determine a core essential region of the genome associated with the 5' end (with a high density of essential genes, labeled in red), and a non-essential one located at the 3' of the genome (labeled in blue) (Fig. 5a). Finally, not all genes present at the core genome are essential since knock out of some of these genes was accomplished (Fig. 5a–c). This essential core region was even smaller in *P. macleodensis*, which lacks a recent genomic inversion present in *P. neocaledonia* (Fig. 5d), located at the intersection of the essential and non-essential zone of the genome (Fig. 5a)[3]. Concordantly, this inversion in *P. neocaledonia* has introduced a fragment of the non-essential region of the genome into a newly fragmented essential core region of its genome (Fig. 5d).

## Genomic organization of pandoravirus conserves traces of their putative evolution from smaller viral ancestors

The distribution of essential genes observed in pandoravirus would be parsimoniously explained by the expanding genome hypothesis[9] and rather hardly with the reductive one[8]. Concordantly, reductive evolution in pathogenic bacteria or eukaryotes did not lead to the concentration of essential genes in particular regions of their genomes[32,33]. To test the genome expansion hypothesis in pandoravirus, we classified genes according to their conservation in their distant relatives within *Nucleocytoviricota* (Fig. 6a)[9,25], and analyzed their position in the genome of *P. neocaledonia* (Fig. 6b). P. neocaledonia genes with orthologs in *Phycodnaviridae* and mollivirus are enriched at the 5' of the genome, coinciding with the essential core of the genome (Fig. 6b). On the other hand, clade-specific genes are more uniformly distributed and strain-specific genes (ORFan genes) tend to accumulate at the 3' end of the genome (Fig. 6b). Concordantly, these data support the hypothesis of a biased expansion of the pandoravirus genome from a *Phycodnaviridae*-like virus towards the 3' end of the genome.

Importantly, gene persistence throughout evolution correlates with the fitness cost associated with the loss of function of that gene[34]. Thus, to further test the genome expansion theory of pandoravirus, we attempted gene KO for approximately 10% of the genes present in the essential region of the genome (Fig. 6c and S4). In agreement with the genome expansion theory, the density of essential genes increased at the positions where *Phycodnaviridae* and mollivirus conserved genes are also highly represented (Fig. 6c). The phenotype of non-essential genes was not associated with the percentage of infectious particles produced but rather the total number of particles produced per cell as shown by the infectious dose 50 (Fig S4c). Moreover, the assessment of the depth of conservation of genes across *Nucleocytoviricota* versus the phenotype associated with their deletion indicates that shared genes have a greater tendency to produce a more drastic effect upon deletion (Fig. 6d). This distribution of phenotypes followed the trend predicted by the genome expansion theory and correlates with the proposed evolutionary pathway from a phycodnavirus-like ancestor to pandoravirus (Fig. 6a). Moreover, genes largely conserved across the *Nucleocytoviricota* (despite multiple independent deletions rendered only one gene strictly conserved in all viruses[35]) that were used for the phylogenetic reconstruction of the phylum are essential for pandoravirus replication (Fig. 6d). These genes include: B DNA Polymerase (PolB), A32-like packaging ATPase (A32), virus late transcription factor 3 (VLTF3), large RNA polymerase subunits (RNAP1 and RNAP2), TFIIB transcriptional factor (TFIIB) and D5-like primase-helicase[9,25,35]. These data align with their use for phylogenetic reconstructions by indicating their likely ancestry for the phylum.

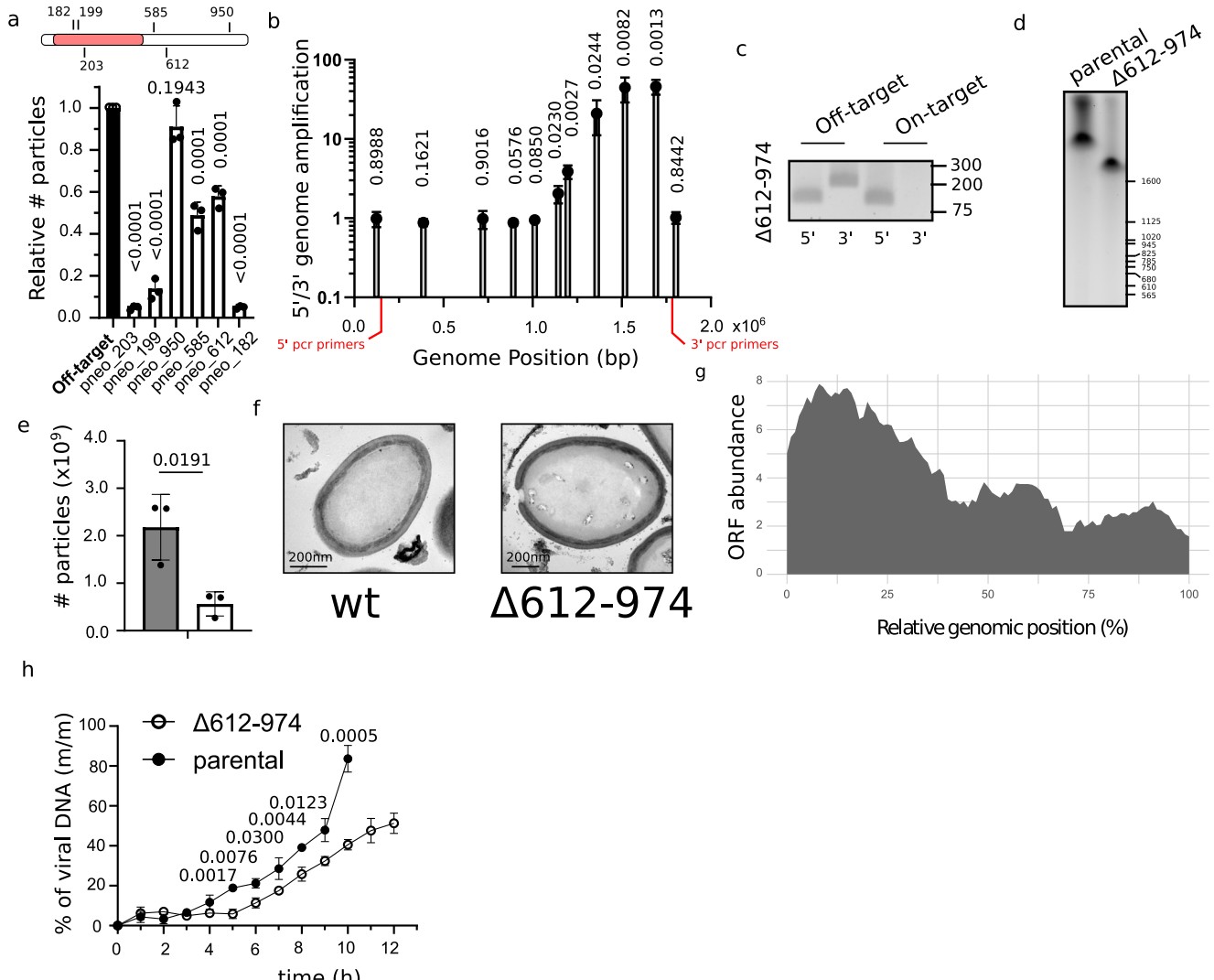

**Fig. 4 | CRISPR/Cas9 targeting of pandoravirus genome demonstrates the presence of a core essential genome. a** Relative quantification of the number (#) of viruses produced 24 hpi of CRISPR/Cas9 expressing amoebas. Data correspond to the mean ± SD of 3 independent experiments. MOI = 1. The essential region of the genome is highlighted in red. Guides targeting mollivirus *mcp* were used as off-target gRNAs. The null hypothesis (α = 0.05) was tested using unpaired two-tailed Student's *t* tests. **b** qPCR analysis indicates the presence of virus with large deletions at the 3′ of the genome. A ratio of the copy number of genes pneo_74 (5′) and pneo_974 (3′) is shown (location marked in red). The location of the bars indicates the position of the targeted gene. Data correspond to the mean ± SD of 3 independent experiments. The null hypothesis (α = 0.05) was tested using unpaired two-tailed Student's *t* tests. **c** Clonality of viral strains with large deletions at the 3′ of the genome was analyzed by PCR. Expected size: 5′: 150 bp (*pneo_74*), 3′: 220 bp (*pneo_974*). PCRs are representative of 2 independent experiments. **d** Pulsed field gel electrophoresis

of the genomic DNA of *Pandoravirus neocaledonia*. Genome obtained from parental virions (2.05 Mbp) or virions containing a large deletion at the 3′ end of the genome (Δ612-974) are shown. Image is representative of 2 independent experiments. **e** Quantification of the number (#) of viruses produced 24 hpi. Data correspond to the mean ± SD of 3 independent experiments. Parental strain and Δ612-974 are represented in gray and while, respectively. The null hypothesis (α = 0.05) was tested using unpaired two-tailed Student's *t* tests. **f** TEM image of an ultrathin section of *P. neocaledonia* viral particle. Wt and Δ612-974 are shown. Micrographs are representative of 3 independent experiments. **g** Abundance of proteins detected in the *P. neocaledonia* virion proteome according to the genomic location of cognate genes. ORF: open reading frame. **h** DNA replication was analyzed by qPCR. Viral DNA is represented as a percentage of total DNA in the sample. Data correspond to the mean ± SD of 3 independent experiments. The null hypothesis (α = 0.05) was tested using unpaired two-tailed Student's *t* tests.

Finally, analysis of gene essentiality according to gene broad functional annotations indicates that essential genes accumulate particularly under "replication" and "transcription" functions, while "Translation and amino-acid metabolism" includes none (Fig. 6e). This observation further supports the viral origin of pandoravirus and indicates that translation-related genes might have been acquired recently to maximize the expression of viral genes[36]. Genes with unknown functions, including all the ORFan genes, englobe a large amount of non-essential genes. Thus, these genes are also likely to be more recent acquisitions of the pandoravirus.

Previous reports have shown an unexpected prevalence of multiple-copy genes in Pandoravirus genomes[3]. Moreover, the speed

of acquisition of these multi-copy genes increased with its putative genome expansion (Fig. 6a). These data suggest that genome expansion was accompanied by a strong increase in genetic redundancy and thus, led to increased robustness of the biological system[37]. To test this hypothesis, we analyzed a family of SNF2-like genes present in *Phycodnaviridae*, mollivirus, and pandoravirus. SNF2 genes are helicase-related ATPases which in eukaryotic cells drive chromatin-remodeling[38]. While *Phycodnaviridae* and mollivirus encode only one SNF2-like gene, this gene is present in two copies in the genome of pandoravirus. Interestingly, phylogenetic analysis of these genes indicates that the duplication occurred in the ancestor of pandoravirus and mollivirus (Fig. S5). A followed-up deletion of one of these genes

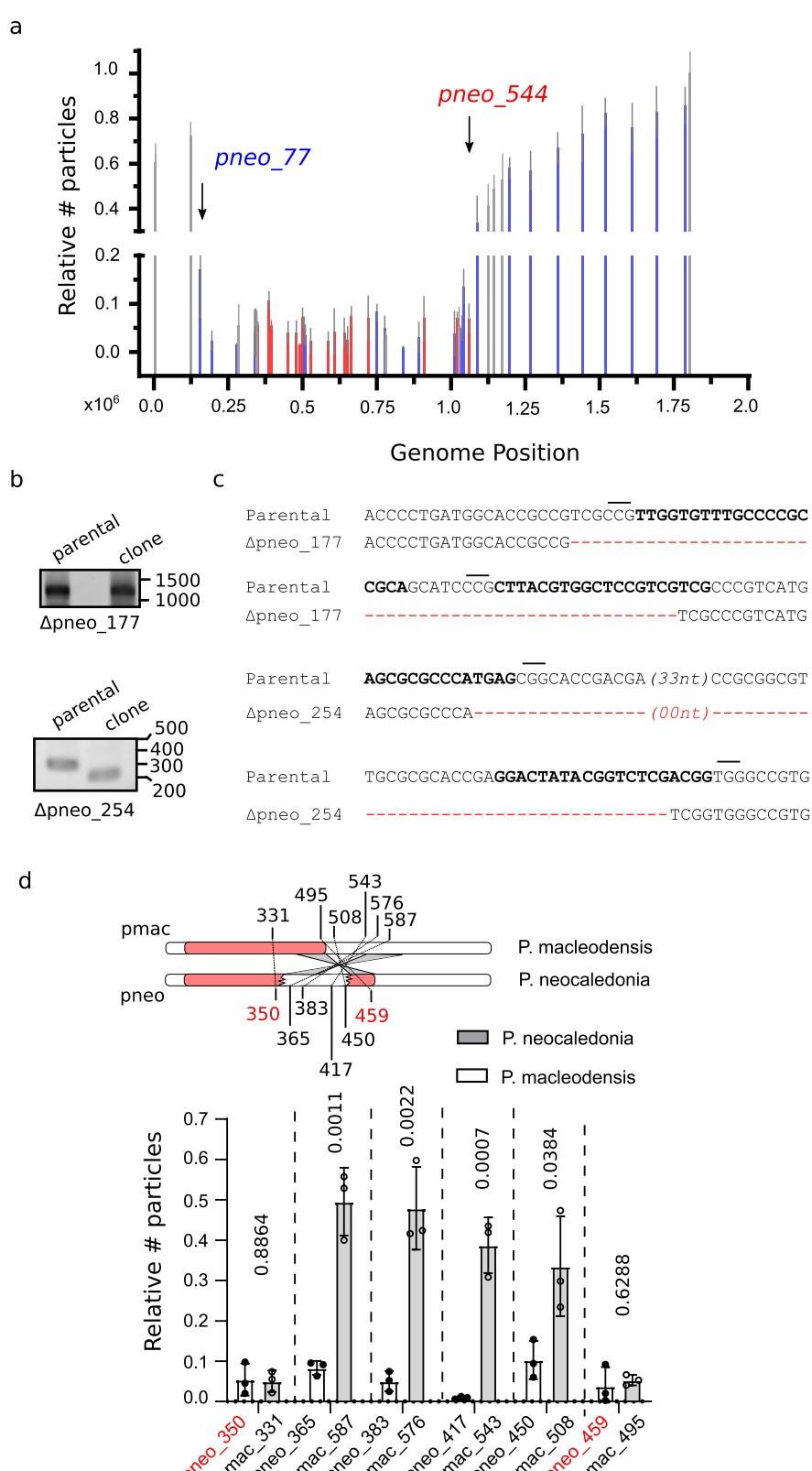

would explain the presence of a single gene in mollivirus (Fig. S5). Such evolutionary history can be explained by the proposed "genomic accordion" evolution proposed for GVs[39,40]. Importantly, both genes present in pandoravirus were shown dispensable individually (Fig. 7a) and viruses encoding a deletion of these genes were easily cloned (Fig. 7b, c). On the other hand, double KO led to synthetic lethality (Fig. 7b, c). Concordantly, these data demonstrate the presence of

epistatic effects and genetic redundancy in GVs. Regardless, both genes are not strictly redundant since a fitness cost associated with single deletion was observed. To assess if genetic redundancy increased with genome expansion, we targeted the SNF2-like gene in mollivirus, which is present in a single copy. We were not able to knockout *mk_236*, which strongly supports its essentiality (Fig. 7a–c). Thus, such redundancy displayed by pandoravirus would indicate that the

**Fig. 5 | CRISPR/Cas9 targeting of pandoravirus genome demonstrates the presence of a core essential genome. a** Quantification of the number (#) of viruses produced 24 hpi of CRISPR/Cas9 expressing amoebas relative to viruses produced in off-target gRNA expressing amoebas. Genes were ranked by their position in the genome. Guides off-target and on-target of different genes were used. Data correspond to the mean ± SD of 3 independent experiments. Genes confirmed to be essential, dispensable or not analyzed are represented in red, blue and gray, respectively. The approximate limits of the essential core of the genome are marked with arrows and the targeted gene indicated. MOI = 1. Guides targeting mollivirus *mcp* were used as off-target gRNAs. **b** Gene disruption of gene of interest (*goi*) is demonstrated by PCR. Expected PCR size: pneo_177: wt, 1250 bp – mutant, 1200 bp; pneo_254: wt, 320 bp – mutant, 220 bp. Cartoon depicting the strategy for genotyping is shown in Fig. S4A. PCRs are representative of 2 independent experiments. **c** Sequencing results of targeted region on clonal populations of Δ*pneo_177* and Δ*pneo_254*. The deleted sequences are shown in red. gRNA sequences are highlighted in bold and PAM sequences marked with a line. PAM for pneo_177 is shown for the reverse complement strain. **d** Quantification of the number (#) of virions of *P. neocaledonia* and *P. macleodensis* produced 24 hpi of CRISPR/Cas9 expressing amoebas relative to viruses produced in off-target gRNA expressing amoebas. Genes were ranked by their position in the genome of p. neocaledonia. Essential genes in *P. neocaledonia* are highlighted in red. Orthologous genes between *P. neocaledonia* and *P. macleodensis* are shown in pairs. A schematic representation of the genome organization of *P. neocaledonia* and *P. macleodensis* including gRNA targeting locus locations is also shown. The limits of the inverted region of the genome is shown in gray. The essential core of the genome is represented with a red box. MOI = 1. Guides targeting mollivirus *mcp* were used as off-target gRNAs. Data correspond to the mean ± SD of 3 independent experiments. Source data are provided as a Source Data file.

flexibility for adaptation to uncertain environments might generate a strong selective pressure in these viruses. Concordantly, increased genetic robustness might be a key component that explains the selective advantage that led to viral giantism.

## Major capsid protein (MCP) is essential for virion formation in mollivirus

Viruses belonging to the *Nucleocytoviricota* tended to lose some core genes independently throughout evolution[9,35]. In particular, pandoravirus lacks a *Nucleocytoviricota* signature MCP protein[9]. MCP is a major component of the viral particles of icosahedral viruses and a key factor for the shaping of their capsids[41]. Interestingly, despite its spherical shape, mollivirus encodes an MCP-like protein[14], and it has been speculated that this gene would be a trace of their icosahedral ancestry (Fig. 8a)[12]. AlphaFold predictions[42] confirmed the jelly-roll folding unit of *ms_334* which closely resembles the MCP of other icosahedral viruses like *Paramecium bursaria* chlorella virus 1 (PBCV-1)[41] (Fig. 8b). To assess the role of this protein in the formation of mollivirus particles, we first expressed an ectopic copy of the gene in the amoeba. Fluorescent N- or C-terminally tagged MCP localized into the cytoplasm of non-infected amoebas but re-localized to specific subcompartments upon infection by mollivirus (Fig. 8c). Moreover, incorporation of MCP to the viral particle was confirmed by fluorescence (Fig. 8d). To attempt genetic deletion of mollivirus MCP, we infected transgenic amoebas expressing Cas9 and gRNAs targeting the *mcp* gene in mollivirus. Targeting of *mcp* efficiently reduced the viral fitness of mollivirus (Fig. 8e). Trans-complementation, as we have previously described[43], was not successful to restore viral growth (Fig. 8f). Expression quantities or timing might explain the failure of trans-complementation. Thus, we developed a system for cis-complementation of gene KO (Fig. 8f). To achieve cis-complementation, we built a plasmid that encodes for a second copy of the *mcp* gene containing silent mutations at the gRNA target sites; and a nourseothricin N-acetyl transferase (NAT) resistance cassette, which confers resistance to nourseothricin[30]. The NAT selection cassette was expressed under the control of a pandoravirus promoter, which can be used for expression of genes in mollivirus (Fig. 8f). Importantly, such recombinant viruses contain two copies of the *mcp*: one susceptible to cleavage by Cas9, and one resistant to its targeting (Fig. 8f and Fig. S5). Concordantly, double-strand breaks would still occur in the cis-complemented strain which might be repaired in order to produce viable progeny. Contrary to trans-complementation, cis-complementation efficiently restored viral fitness upon Cas9 cleavage of *mcp* (Fig. 8e). Thse data demonstrate that defects in the viral cycle associated with Cas9 cleavage are due to the knockout of the targeted gene rather than a general response to double-strand breaks. Moreover, while targeting of *mcp* did not impact DNA replication of the virus (Fig. 8g), biogenesis of virions was largely impaired (Fig. 8h). In contrast to untargeted or cis-complemented viruses, targeting of *mcp* on wild-type viruses led to the accumulation of membranous

material in the cytoplasm and absence of mature neosynthesized virions (Fig. 8h). These data indicate a failure to initiate tegument biogenesis upon recruitment of an open cisterna[44] and suggests a scaffolding function for mollivirus MCP. Overall, these data support a replacement of the MCP functions during the evolution of phycodnavirus-like icosahedral viruses to mollivirus-like.

## Discussion

The study of the origin of viruses and their evolutionary history remains rather challenging for a reliable phylogenetic reconstruction due to their fast evolution and quick exchange of genetic markers[9,45]. Regardless, such fast evolution in dsDNA viruses (with low mutation rates but high burst size[46]) might have been key to the evolution of all living cells[47,48], and would have contributed to shaping modern genes[25], genetic elements[49], or even organelles like the nucleus[50,51]. Here, we introduced a battery of tools that allows genetic manipulation of host and nuclear GV and thus, the trackability of these gene exchanges. This includes the characterization of the neglected functional genomic of nuclear GV and their up to 70% ORFan gene-containing genomes. ORFan genes have contributed substantially as evolutionary innovations[4] and are expected to arise as lineage-specific adaptations to the environment[52,53]. The role of such innovations and unpredictable functions in pandoravirus are now open to rigorous investigation.

Particularly, we demonstrate the existence of an essential core of the genome located at the 5´ end of pandoravirus genome (Fig. 5a), which coincides with the location of *Phycodnaviridae* and mollivirus-conserved ancestral genes (Fig. 6b). These data, together with the phylogeny recently accepted by the International Committee on Taxonomy of Viruses[54], support the hypothesis that nuclear GVs likely originated from smaller icosahedral viruses[36]. Moreover, the presence of an icosahedral *mcp* gene in mollivirus, which is essential for virion formation despite its ovoid shape further supports this theory. If pandoravirus ancestors possessed mollivirus-like particles and which molecular changes allowed the dispensability of the scaffolding functions of MCP are yet to be studied.

Interestingly, we also show that while discrete mutations can easily be isolated when the 5′ of the genome is targeted by CRISPR/ Cas9, large deletions are frequent when the 3′ of the genome is targeted. These differences might be due to several reasons. First, the repair mechanism might not defer across the genome but the high density of essential genes at the 5′ of the genome could select for the viruses with small deletions. Second, DNA compaction and transcription highly affect Cas9 targeting in mammalian cells[55]. Differences in density of late versus early genes at different positions of the genome might also indicate differences in DNA compaction or transcription activity along the viral infectious cycle. Third, it has been previously shown for herpesvirus that replication of the DNA influences the insertion and deletion lengths[56]. Moreover, replication of the DNA in vaccinia virus has been shown to initiate near the end of the genome[57].

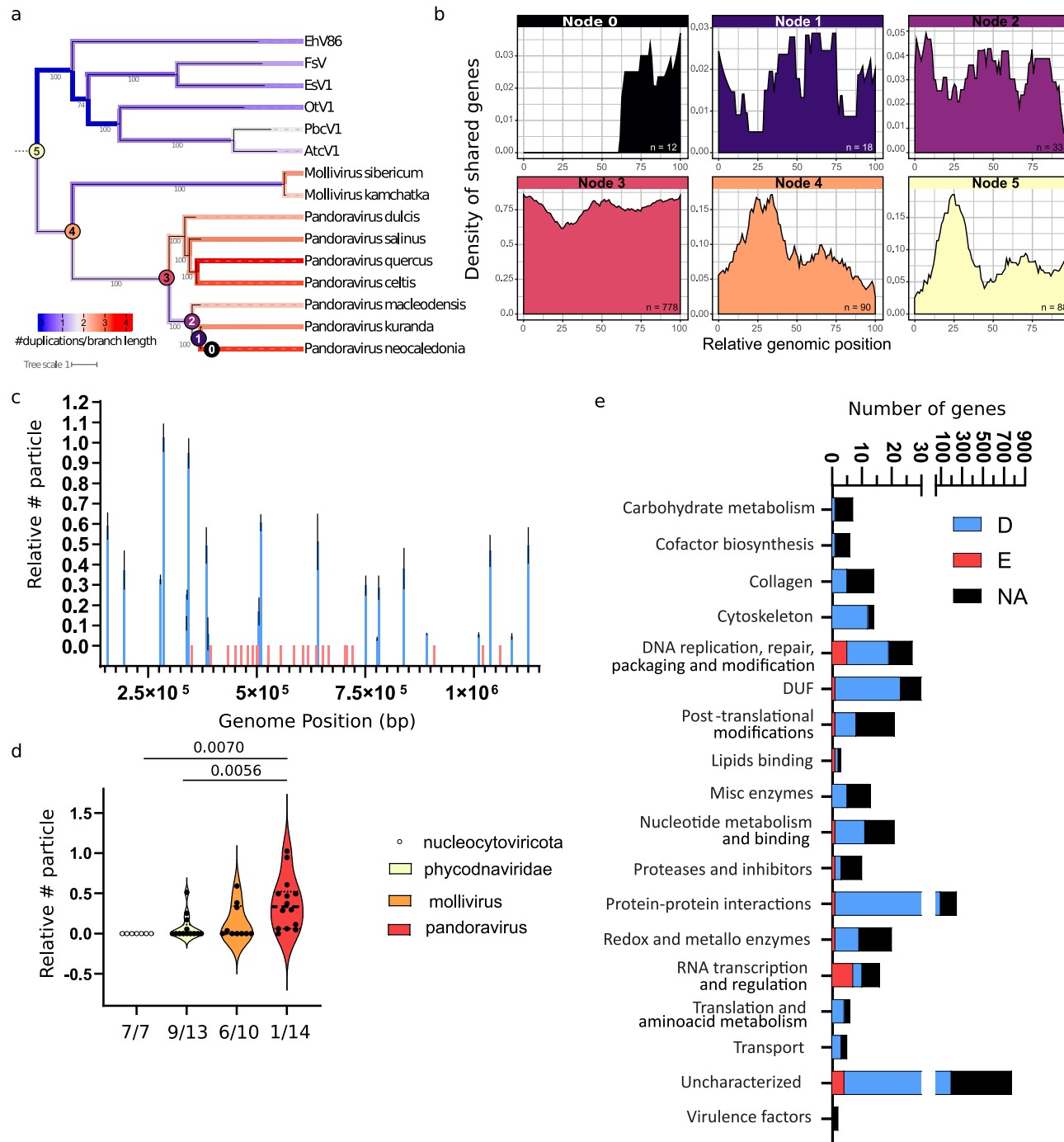

**Fig. 6 | Pandoravirus genome organization keeps traces of their smaller ancestors. a** Phylogenetic tree of *Phycodnaviridae*, *Molliviridae* and *Pandoraviridae* computed from a concatenated multiple alignment of 435 orthologous proteins present in at least two viruses. Bootstrap values were computed using the ultrafast boostrap method[64]. Number of gene duplication events along the branches from the phylogenetic tree are also shown. The numbers were normalized by branch length, log10 transformed and color-coded from blue (low number of duplications) to red (high). **b** Density of *P. neocaledonia* genes along the genome according to its predicted ancestry. The *P. neocaledonia* genes were classified based on the presence of homologs in other viruses and color-coded from the corresponding ancestral node shown in panel **a**. **c** Quantification of the number (#) of viruses produced upon knock-out of genes located at the essential core of the genome relative to viruses produced in off-target gRNA expressing amoebas and ranked by their position in the genome. Data correspond to the mean ± SD of 3 independent experiments. Genes that were knocked out are highlighted in blue, while genes that could not be knocked out are shown in red. All experiments were performed with clonal populations of recombinant viruses. MOI = 1. **d** Correlation between the number of particles produced 24 hpi upon single gene knock out and proposed depth of conservation. The phylogenetic relationship of *P. neocaledonia* with other *Nucleocytoviricota* is illustrated in Fig. 4a. The distribution of phenotype scores in each category is plotted and the number of genes analyzed for each category is showed on the bottom of each violin plot. Bars indicate the group median. The number of essential genes over the total of genes analyzed in each group is also indicated. MOI = 1. The null hypothesis (α = 0.05) was tested using unpaired two-tailed Student's *t* tests. **e** Correlation between dispensability/essentiality (D = dispensable, E = essential, NA = not analyzed) and functional annotation of *Pandoravirus neocaledonia* genes.

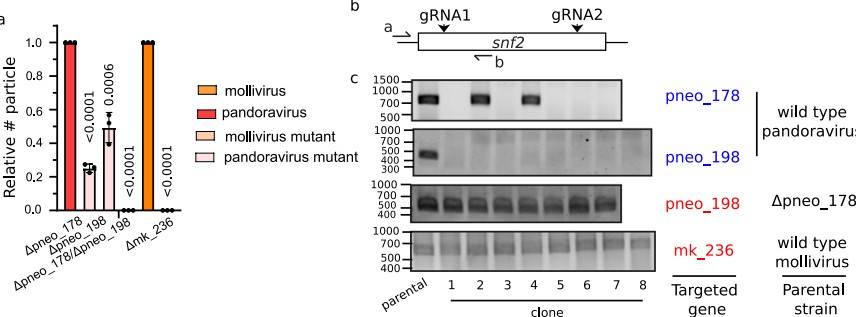

**Fig. 7 | Pandoravirus genome displays gene redundancy. a** Quantification of the number of viruses produced upon knock-out of *snf2* like genes in pandoravirus or mollivirus relative to viruses produced in off-target gRNA expressing amoebas. Data correspond to the mean ± SD of 3 independent experiments. MOI = 1. The null hypothesis (α = 0.05) was tested using unpaired two-tailed Student's *t* tests. **b** Cartoon depicting the strategy for genotyping shown in (Fig. 4h). **c** Gene presence or disruption of *goi* is demonstrated by PCR in different clones obtained upon gene targeting by Cas9. PCR positive clones represent viruses which escape Cas9 targeting. Expected PCR product size: pneo_178: 790 bp; pneo_198: 466 bp; mk_236: 562 bp. Source data are provided as a Source Data file.

While the location of the origin of replication of pandoravirus is unknown, if such origin would be located at the extremities of the genome, it might influence repair efficiencies similarly to what was described for herpesvirus. However, the exact mechanism governing these differences in different regions of the viral genome remains to be studied.

Overall, we anticipate that the development of reverse genetic approaches in the field of GVs will lead to a vast expansion of the knowledge about these outliers of the viral world and shed light on the intricated relationship with their cellular hosts.

## Methods

### Viral strains utilized in this work

The following viral strains have been used in this study: *Pandoravirus neocaledonia*[3], *Pandoravirus macleodensis*[3], *Pandoravirus kuranda* (this study, ON887157), *Mollivirus kamchatka*[26], *Pithovirus sibericum*[27] and *Mimivirus reunion*[28].

### Cloning of DNA constructs

All primers used in this study are listed in Table S1. All vectors used in this study are listed in Table S2.

**CRISPR/Cas9 expression vector.** Vectors for CRISPR/Cas9 expression in *A. castellanii* cells were generated by inserting a codon optimized *Sp*Cas9 (with or without a nuclear localization signal (NLS)) and a cassette for polycistronic gRNA expression into the pEF1-GFP-NEO vector[43]. The cassette for polycistronic gRNA expression was synthesized by GenScript and designed to contain an *A. castellanii* U6 promoter-tRNA-gRNA1-tracerRNA1-tRNA-gRNA2-tracerRNA2. This cassette was further modified by PCR and circularized by recombination (InFusion Takara) to replace gRNA1-tracerRNA1-gRNA2 by a NotI site. gRNAs were included in primers synthesized by eurofins genomics, and inserted into the NotI site by InFusion (Takara).

**MCP-mRFP and mRFP-MCP expression vectors.** PCR product of the *mcp* was generated using *M. kamchatka* genomic DNA produced by Wizard genomic DNA purification kit (PROMEGA) according to manufacturer's specifications. Primers used are HB84/HB85 and HB86/HB87. Vectors for expression of *mcp* were generated using the pEF1-mRFP-NAT vector by InFusion (Takara) into the NdeI or XhoI site for N- or C-terminal RFP tagging, respectively.

**MCP second copy.** The plasmid for cis-complementation was generated by sequential cloning of the 3' UTR of pneo_480, the promoter of pneo_650, a Nourseothricin N-acetyl transferase (NAT) selection cassette, the 3' UTR of pneo_650, the promoter of *mcp* and the *mcp* coding sequence. Each cloning step was performed using the Phusion Taq polymerase (ThermoFisher) and InFusion (Takara). Silent mutations at the gRNA targeting site were also introduced using InFusion. Finally, 500 bp homology arms were introduced at the 5' and 3' end of the cassette in order to induce homologous recombination with the viral DNA. Prior to transfection, plasmids were digested with EcoRI and NotI.

**Endogenous tagging**

The plasmid for endogenous tagging was generated by sequential cloning of the 3xHA/FLAG tag, 3' UTR of pneo_480, the promoter of pneo_650, a Nourseothricin N-acetyl transferase (NAT) selection cassette, the 3' UTR of pneo_650. Each cloning step was performed using the Phusion Taq polymerase (ThermoFisher) and InFusion (Takara). Finally, 500 bp homology arms were introduced at the 5' and 3' end of the cassette in order to induce homologous recombination with the viral DNA. Prior to transfection, plasmids were digested with EcoRI and NotI.

**cs1 (cellulose synthase) second copy**

The *cs1* gene (XP_004335167.1) was amplified by PCR from genomic DNA and cloned into a pGAPDH-GFP amoebal expression plasmid[58]. The plasmid was linearized by NdeI and the gene was inserted in the plasmid Vc2[43] used for recombination (InFusion Takara).

### Cell culture and establishment of cell lines

**Cell culture.** *Acanthamoeba castellanii* (Douglas) Neff (American Type Culture Collection 30010TM) cells were cultured at 32 °C in 2% (wt/vol) proteose peptone, 0.1% yeast extract, 100 μM glucose, 4 mM MgSO$_4$, 0.4 mM CaCl$_2$, 50 μM Fe(NH$_4$)$_2$(SO$_4$)$_2$, 2.5 mM Na$_2$HPO$_4$, 2.5 mM KH$_2$PO$_4$, pH 6.5 (home-made PPYG) medium supplemented with antibiotics [ampicilline 100 μg/mL, and Kanamycin 25 μg/mL]. 100 μg/mL Geneticin G418 or Nourseothricin was added when necessary.

**Cell transfection.** 1.5×10$^5$ *Acanthamoeba castellanii* cells were transfected with 6 μg of each plasmid using Polyfect (QIAGEN) in phosphate saline buffer (PBS) according to the manufacturer's instructions. Selection of transformed cells was initially performed at 30 μg/mL Geneticin G418 or Nourseothricin and increased up to 100 μg/mL within a couple of weeks.

### Establishment of viral lines

**Cas9 KO viral lines.** Cell lines expressing Cas9 and gRNA were infected with a MOI (Multiplicity Of Infection) of 1. Infection was allowed to proceed for 24 h. This new generation of viruses was used to quantify

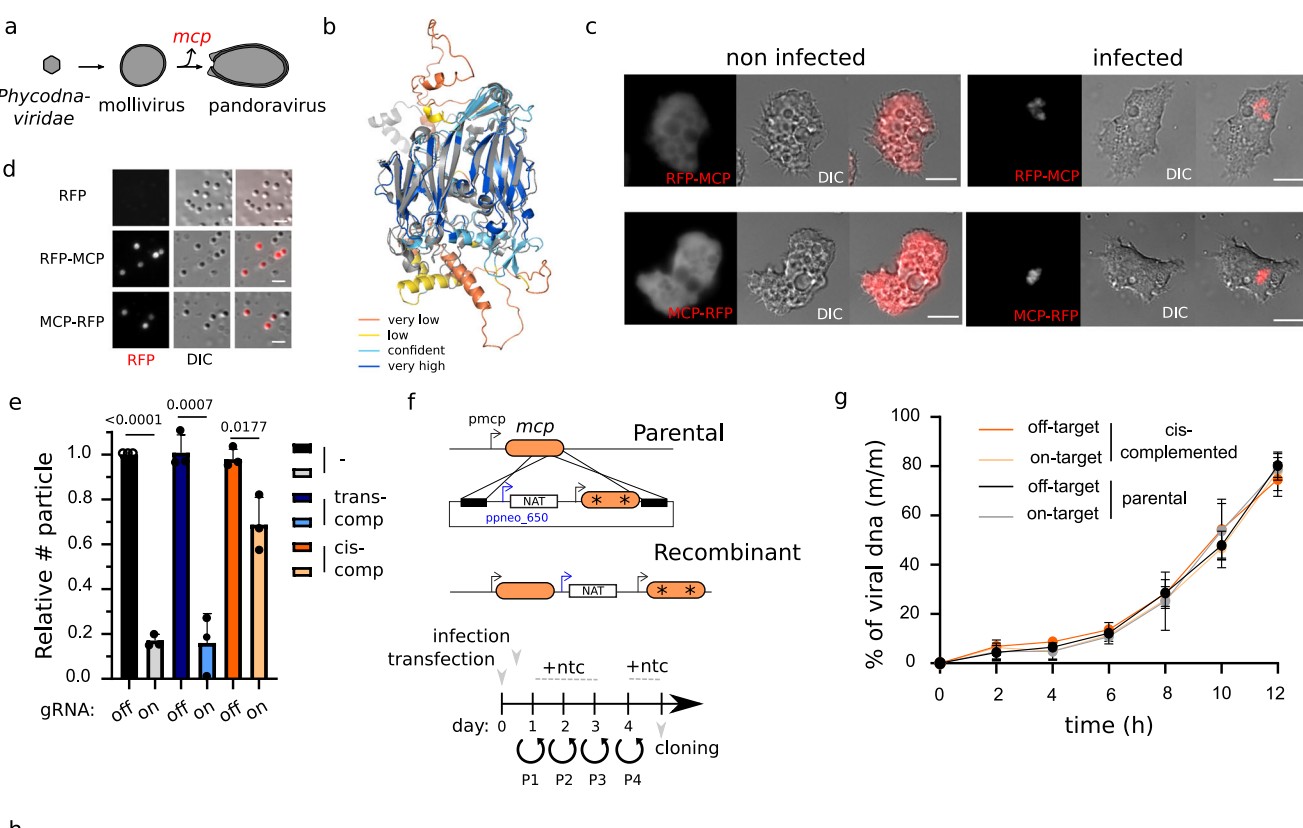

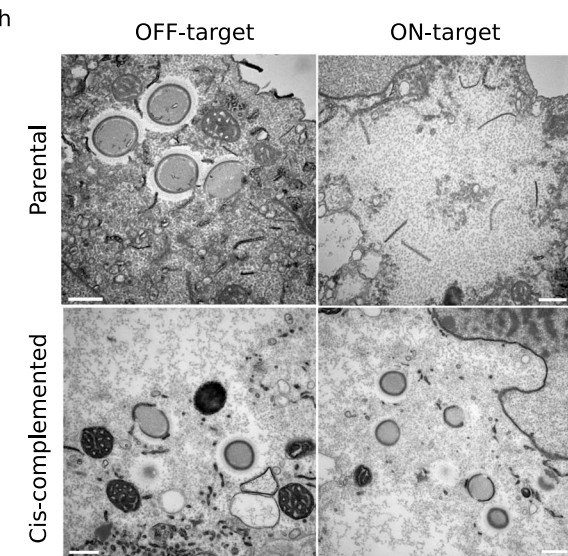

**Fig. 8 | Mollivirus MCP acts as a scaffolding protein for tegument biosynthesis.**
**a** Schematic depiction of the proposed model of evolution for pandoravirus. Major capsid protein loss is indicated by a red color. **b** Alignment of the AlphaFold prediction of the mollivirus major capsid protein (mcp, ms_334) and the crystal structure of the PBCV-1 mcp (5TIQ). PBCV-1 mcp is shown in gray while mollivirus mcp is colored by its confidence score (pLDDT): Very low (0<pLDDT<50), low (50<pLDDT<70), confident (70<pLDDT<90) and high (90<pLDDT<100). **c** Light fluorescence microscopy images of *A. castellanii* cells expressing N- or C-terminally RFP tagged MCP. Representative images of non-infected cells and cells infected with mollivirus are shown. Images were obtained 6 hpi. Scale bar: 10μm. **d** Light fluorescence microscopy images of mollivirus produced in *A. castellanii* cells expressing RFP or N- or C-terminally RFP tagged MCP. Scale bar: 1 μm.
**e** Quantification of the number of viruses produced upon knock-out of *mcp* gene in mollivirus relative to viruses produced in off-target gRNA expressing amoebas. Data correspond to the mean ± SD of 3 independent experiments. * indicates the presence of silent mutations on the sequence. MOI = 1. Guides targeting pandoravirus *rpb1* were used as off-target gRNAs. The null hypothesis (α = 0.05) was tested using unpaired two-tailed Student's *t* tests. **f** Schematic representation of the vector and strategy utilized for *mcp* KO cis-complementation. Selection cassette was introduced by homologous recombination and recombinant viruses were generated, selected and cloned as indicated by the timeline. Viral infection was performed 1 hpi. Ntc: Nourseothricin. * represents the presence of silent mutations at the gRNA targeting site. **g** DNA replication was analyzed by qPCR. Viral DNA is represented as a percentage of total DNA in the sample. Data correspond to the mean ± SD of 3 independent experiments. MOI = 1. **h** Electron microscopy imaging of the mollivirus replication cycle in *A. castellanii*. Images were acquired 8 hpi and mollivirus particles (MOI = 50) were used to infect cells expressing Cas9 and gRNA on- or off-target of the *mcp*. Parental or cis-complemented viral strains were used for infection in parallel experiments. Scale bar: 500 nm. Source data are provided as a Source Data file.

the number of particles produced or cloned for isolation of viral strains.

## Cas9 KO cell lines for cellulose synthase

$1.5 \times 10^5$ *Acanthamoeba castellanii* cells were transfected with 6 μg of the plasmid expressing Cas9 and gRNAs against the *cs1-3* using Polyfect (QIAGEN) in phosphate saline buffer (PBS). Selection of transformed cells was initially performed at 30 μg/mL Geneticin G418. Upon recovery of the cells, amoeba culture was cloned and genotyped.

*Insertion of MCP second copy and endogenous tagging.* $1.5 \times 10^5$ *Acanthamoeba castellanii* cells were transfected with 6 μg of linearized plasmid using Polyfect (QIAGEN) in phosphate saline buffer (PBS). One hour after transfection, PBS was replaced with PPYG and cells were infected with $1.5 \times 10^7$ mollivirus particles for 1 h with sequential washes to remove extracellular virions. 24 h after infection the new generation of viruses (P0) was collected and used to infect new cells. An aliquot of P0 viruses was utilized for genotyping in order to confirm integration of the selection cassette. New infection was allowed to proceed for 1 h, then washed to remove extracellular virions and nourseothricin was added to the media. Viral growth was allowed to proceed for 24 h. This procedure was repeated one more time before removing the nourseothricin selection to allow viruses to expand more rapidly. Once, viral infection was visible, selection procedure was repeated one more time. Viruses produced after this new round of selection were used for genotyping and cloning.

*Cloning and genotyping.* 150,000 *A. castellanii* cells were seeded on 6 wells plates with 2 mL of PPYG. After adhesion, viruses were added to the well at a multiplicity of infection (MOI) of 2. One hour post-infection, the well was washed 5 times with 1 mL of PPYG and cells were recovered by scraping. Amoebas were then diluted until obtaining a suspension of 1 amoeba/μL. One μL of such suspension was added in each well of a 96-well plate containing 1000 uninfected *A. castellanii* cells and 200 μL of PPYG. Wells were later monitored for cell death and 100 μL collected for genotyping. Genotyping was performed using Terra PCR Direct Polymerase Mix (Takara) following manufacturers specifications.

*Virus Purification.* The wells presenting a recombinant genotyping were recovered, centrifuged 5 min at 500 × g to remove the cellular debris, and used to infect ten 75 cm² tissue-culture flasks plated with fresh *Acanthamoeba* cells. After lysis completion, the cultures were recovered, centrifuged 5 min at 500 × g to remove the cellular debris, and the virus was pelleted by a 45 min centrifugation at 6800 × g prior purification. The viral pellet was then resuspended and washed twice in PBS and layered on a discontinuous CsCl gradient (1.2/1.3/1.4/1.5 g/cm³), and centrifuged at 100,000 × g overnight. An extended protocol is shown in ref. [59].

## Fluorescence imaging

Transfected *A. castellanii* cells were grown on poly-L-lysine coated coverslips in a 12-well plate and fixed with PBS containing 3.7% formaldehyde for 20 min at room temperature. After one wash with PBS buffer, coverslips were mounted on a glass slide with 4 μl of VEC-TASHIELD mounting medium with DAPI and the fluorescence was observed using a Zeiss Axio Observer Z1 inverted microscope using a 63x objective lens associated with a 1.6x Optovar for DIC, mRFP or GFP fluorescence recording. Quantification of fluorescence was performed in at least 200 cells from no less than 10 different fields of view.

## Amoeba encystation and quantification by SDS resistance

Acanthamoeba cysts were induced in encystation media (100 mM KCl, 80 mM MgSO₄, 0.4 mM CaCl₂, and 20 mM 2-amino-2-methyl-1,3-propanediol, pH 9.0) for 3 days, as previously reported[23]. Cysts were counted under a light microscope upon treatment with 0.5% SDS for 10 minutes and encystation ratios were calculated[23].

## Calcofluor white staining

Calcofluor white staining was performed as previously described[23]. Briefly, *A castellanii* cells were incubated in encystation media for 3 days and amoeba suspensions were incubated for 20 minutes at room temperature with 2.5% calcofluor white staining solution. After washing with PBS, samples were observed under a fluorescence microscope.

## Indirect immunofluorescence analysis (IFA)

*A. castellanii* cells were grown on 96-well plates and infected with pandoravirus (MOI = 10). Cells were fixed at 6 hpi (post-infection) using 4% paraformaldehyde. Immunofluorescence analysis was performed as previously described[60]. Nuclei were stained with Hoescht, and coverslips were mounted in Fluoromount G (Southern Biotech).

## Infective dose 50

ID50 were calculated as previously described[61]. In brief, particle numbers were calculated and adjusted in order to obtain a stock solution of viruses containing $1 \times 10^7$ particles/mL. Serial dilution of this stock was used to infect a 96 wells plate containing 1000 *A. castellanii* per well. Cytopathic effect was then observed one-week post-infection. The end point was later calculated and expressed as 50% infective dose (ID50).

## Statistics and reproducibility

All data are presented as the mean ± s.d. of 3 independent biological replicates ($n$ = 3), unless otherwise stated in the figure. All data analysis were carried out using Graphpad Prism. The null hypothesis ($\alpha$ = 0.05) was tested using unpaired two-tailed Student's *t* tests.

## Viral fitness determination

Viral burst was calculated by manual counting using a hemocytometer chamber. In the case of purified viral samples, optical density was utilized for viral quantification. The purity of the viral samples were analyzed by microscopy as previously described[59].

## Pulsed-field gel electrophoresis

Viral suspensions were prepared according to ref. [62]. Drops of 45 μL of the viral suspension were embedded in 1% low melting agarose, and the plugs were incubated in lysis buffer (50 mM Tris-HCl pH 8.0, 50 mM EDTA, 1% (v/v) laurylsarcosine, and 1 mg/mL proteinase K) for 24 h at 50 °C with light shaking (500 rpm). The lysis buffer was renewed every 8 h and 1 mM DTT was added 30 min before the second buffer change. After lysis, the plugs were washed once in sterile water and three times in TE buffer (10 mM Tris HCl pH 8.0 and 1 mM EDTA), for 15 min at 50 °C. Electrophoresis was carried out in 0.5× TBE using a 1% agarose gel for 20 h 10 min at 6 V/cm, 120° included angle and 14 °C constant temperature in a CHEF-MAPPER system (Bio-Rad) with pulsed times ranging from 0.5 s to 3 min 10 s with a linear ramping factor.

## Quantitative PCR analysis

Viral genomes or gDNA from infected amoebas were purified using Wizard genomic DNA purification kit (PROMEGA). To determine the amplification kinetic, the fluorescence of the EvaGreen dye incorporated into the PCR product was measured at the end of each cycle using SoFast EvaGreen Supermix 2× kit (Bio-Rad, France). A standard curve using gDNA of purified viruses was performed in parallel of each experiment. For each point, a technical triplicate was performed. Quantitative real-time PCR (qRT-PCR) analyses were performed on a CFX96 Real-Time System (Bio-Rad).

## Electron microscopy

Extracellular virions or *A. castellanii*-infected cell cultures were fixed by adding an equal volume of PBS with 2% glutaraldehyde and 20 min incubation at room temperature. Cells were recovered and pelleted 20 min at 5000 × g. Virions were recovered and pelleted 45 min at

$6800 \times g$. The pellet was resuspended in 1 mL PBS with 1% glutaraldehyde, incubated at least 1 h at 4 °C, and washed twice in PBS prior coating in agarose and embedding in Epon resin. Each pellet was mixed with 2% low melting agarose and centrifuged in microvettes tubes (Sarstedt), the agarose coated sample was recovered and cutted to obtain small flanges of approximately 1 mm³. These samples were then prepared using the osmium-thiocarbohydrazide-osmium method: 1 h fixation in 2% osmium tetroxide with 1.5% potassium ferrocyanide, 20 min in 1% thiocarbohydrazide, 30 min in 2% osmium tetroxide, overnight incubation in 1% uranyl acetate, 30 min in lead aspartate, dehydration in increasing ethanol concentrations (50, 70, 90 and 100% ethanol) and embedding in Epon-812. Ultrathin sections of 70 nm were observed using a FEI Tecnai G2 operating at 200 kV[63].

## Comparative genomics and phylogeny

The protein-coding genes annotated in the reassembled *Pandoravirus neocaledonia* genome were compared to the ones of the following viral genomes: *Emiliania huxleyi* virus 86 (NC_007346.1), Feldmannia species virus (NC_011183.1), *Ectocarpus siliculosus* virus 1 (NC_002687.1), *Ostreococcus tauri* virus 1 (NC_013288.1), *Paramecium bursaria* Chlorella virus 1 (NC_000852.5), *Mollivirus sibericum* (NC_027867.1), *Mollivirus kamchatka* (MN812837.1), *Pandoravirus dulcis* (NC_021858.1), *Pandoravirus salinus* (NC_022098.1), *Pandoravirus quercus* (NC_037667.1), *Pandoravirus macleodensis* (NC_037665.1) and *Pandoravirus kuranda* (ON887157, this study, https://www.ncbi.nlm.nih.gov/nuccore/ON887157). We used OrthoFinder (version 2.5.4) to compute the orthogroups (OG) of shared genes using the following parameters: "-M msa -S diamond_ultra_sens -y". From a set of 438 OG selected by OrthoFinder ("Orthogroups_for_concatenated_alignment.txt" file) we kept the 435 OG without paralog, performed a multiple alignment of each OG using Clustal omega (version 1.2.4) and concatenated the multiple alignments using Catsequence. Next a partitioned phylogeny was computed using IQ-Tree (version 2.1.2) with the "-bb 1000 -m MFP + MERGE -rcluster 10" parameters. The root of the tree was calculated using the midpoint rooting method.

## Density of shared genes along the Pandoravirus neocaledonia genome

Each Pandoravirus neocaledonia gene was given an "ancestry score" based on the OGs and the phylogenetic tree. This score indicates the node of the last common ancestor of all viruses having a homologue of the given gene. It ranges from 0 (genes only present in Pandoravirus neocaledonia) up to 5 (ancient, shared with at least one phycodnavirus). We next sliced the Pandoravirus neocaledonia genome in 100 bins and computed the fraction of genes of each category. Finally, these values were smoothed by computing the average values over a sliding window of 20 bins.

## Reporting summary

Further information on research design is available in the Nature Portfolio Reporting Summary linked to this article.

# Data availability

The authors declare that the data supporting the findings of this study are available within the paper and its supplementary information files. Source data are provided with this paper. Plasmids generated in this study can be obtenaid from Addgene or upon request to the authors. Viral strains can be obtained upon request to the authors. Viral genomes: Pandoravirus kuranda (ON887157) Source data are provided with this paper.

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

## Acknowledgements

The authors thank Jean-Michel Claverie for providing the samples to isolate Pandoravirus kuranda and for critical discussion to improve the manuscript. We thank the PiCSL-FBI core facility (Nicolas Brouilly, Fabrice Richard, and Aïcha Aouane, IBDM, AMU-Marseille) and the Plate-forme Transcriptome-IMM/CNRS (Yann Denis). This study was founded by the European Research Council (ERC) under the European Union's Horizon 2020 research and innovation program (grant agreement No 832601; N.P., S.J., and C.A.). H.B. is the recipient of an EMBO Long-Term Fellowship (ALTF 979-2019).

## Author contributions

H.B. contributed to conceptualization, methodology, validation, formal analysis, investigation, visualization and writing of the original draft. M.L. contributed to conceptualization, methodology, software, validation, formal analysis, investigation, visualization and writing (review &

visualization). C.G. contributed to investigation and writing (review & visualization). N.P. contributed to methodology, investigation and writing (review & visualization). J.M.A. contributed to methodology and investigation. S.J. contributed to methodology, investigation and writing (review & visualization). C.A. contributed to conceptualization, project administration, funding acquisition and writing of the original draft.

## Competing interests

The authors declare no competing interests.
