## [Peer Review File · Nature Communications]

Reviewer comments, first round –

Reviewer #1 (Remarks to the Author):

In this manuscript Bision, Legendre, co-authors and Abergel describe the use of CRISPR/Cas9 as a genomic editing tool applied to amoebal giant viruses, particularly those with nuclear phase. This study is original and certainly will inaugurate a new age on giant viruses basic studies. The application of this tool was well explored in the context of pandoravirus biology. A substantial number of constructions and experiments pointed to the main essential spots in pandoravirus genome.

Major points:

- As this work will open the windows to gene function studies on giant viruses field, it would be essential the authors provide a detailed description of materials and methods related to genomic manipulation. It would be very useful a supplemental document containing the sequences of modified GFP and its promoter, RFP, selection markers, etc. I am aware that all this information is precious and authors will apply this tool to future studies. But I see genome editing as the most consistent novelty of this work, therefore details on how to do that would be essential to grant reproducibility.
- Conclusions on pandoravirus and mollivirus evolution sound very speculative. I agree that the position of some clusters of genes would suggest common evolutive features and ancestrality. But even considering all results together the story sounds risky and needs more data: eg. discovery of new viruses to fulfill some gaps. The story of the manuscript don't need to rely on evolution of pandoravirus, because the use of CRISPR/Cas9 to edit giant viruses is a breakthrough to the field.
- In several experiments in this work, particles counting was associated to virus fitness. Could authors explain why do they quantified total particles instead quantify infectious particles? I believe that the measure of infectious particles is specially important in the context of generation of recombination viruses. Some conclusions possibly were based considering a mix of infectious and defective particles. I see the generation of defective particles as a negative factor for viral fitness.

Minor points:

- abstract: is the "fourth domain of life" worth of note? This theory has been refuted by several comprehensive evolutionary studies.
- line 41: metabolism is a generic term. Please, consider specify.
- lines 43-44: "blurred the division between cellular living organisms and viruses". This sentence sounds exaggerated. The authors group published papers on the discovery of some of the most awesome giant viruses ever... and those entities were described as viruses since the beginning by them. I mean, with good sequencing and cycle imaging it is possible to differentiate a giant virus from a cellular organism: FtsZ genes, ribosomes r and proteins, capsid and hallmark genes, eclipse phase, etc, analyzed together in a context.
- line 47: is it necessary to get some federal/local authorization to manipulate the genome of a given organism in France? In some countries this is necessary. If yes, please consider to provide the authorization number.
- Fig 1C: please provide more details on how the fluorescence was measured.
- line 64: Medusavirus does not encode to a RNA pol.
- Fig S1F: why black and white figures?

Reviewer #2 (Remarks to the Author):

Bisio and co-authors presents new methods to use CRISPR/Cas9 to edit giant pandoravirus genomes and study roles of targeted regions in virus replication. While there has been considerable interest in the lifecycle of these megabase genome size viruses, genetic tools have been lacking to enable their manipulation by CRISPR/Cas9 technology. The authors introduce *S. pyogenes* Cas9 together with pU6-driven guide RNAs (gRNA). They show that they are able to edit a model RFP gene in amoeba by CRISPR, and that they can employ a chain reaction approach to increase homologous recombination (though this is not well described). They then test effect on GV that replicate in the nucleus vs cytoplasm, and find that the former are targetable by CRISPR/Cas9. CRISPR approaches are then used to target viral genes, to interrogate their essentiality in viral replication. Effects on viral replication were stronger for gRNA targeting the 5' end of the genome and inducing NHEJ repair, whereas targeting of the gene poor 3' viral genome had comparatively little effects on fitness, despite inducing in some cases large deletions. This interesting difference is not well described at the molecular level – why would targeting of the 3' region result in large deletions, based on understanding of the viral lytic replication mechanisms? The authors then consider pandoravirus genome organization, gene density, and effects of targeting different genomic regions/genes located in the essential core of the genome. Finally, CRISPR is used to test effects of MCP KO on tegument biosynthesis and virus replication.

Overall, the manuscript represents an interesting application of CRISPR/Cas9 technology to studies of GV replication. The authors present strong evidence to support the ability of CRISPR/Cas9 to edit GV genomes by HDR or NHEJ. However, there are a number of issues that would need to be addressed to increase robustness and readability of the manuscript, particularly for a general audience.

Major points:

There is jargon that is not defined and would be best avoided. This makes the manuscript difficult to read. A good example is the term Selfish gene in Fig 1C, which is never defined in the main text or legend. What does this mean? Would take out this term and instead specifically define what is being tested. As another important example, the term mutagenic chain reaction is used in line 57, with no explanation of what this means. This is confusing. This should be much more clearly explained in the text and legend, and not left to reference 15. Likewise, the terms off target and on target are used frequently. Instead of using these potentially confusing terms, define in the text/legend what the "off-target" sgRNAs are actually targeting. You want to use a good control guide that has on-target effects on a control site. And, an on-target sgRNA can have off-target effects elsewhere. I would instead re-name the on-target sgRNA as anti-RFP or whatever it is designed to target, and the off-target sgRNA to indicate its on-target site that allows it to serve as a control.

Since RFP is heavily used for foundational CRISPR experiments in Figure 1, it should be clearly defined in the main text how RFP is expressed in the amoeba. It is said to be episomal, but how many copies is the episome? What is the episome that is used? Why did the authors chose to target an episomal gene and not an amoeba gene for which they have an antibody?

Figure 1C is hard to follow and should be better explained. Why are 100% of cells RFP+ but GFP- in column 1? Does this mean they were not transfected? Then, in column 3, why are some cells GFP- and RFP-? This is confusing. If they are RFP-, presumably the CRISPR worked. But why then did they lose the GFP-Cas9 signal? Likewise, why aren't all of the cells GFP+/RFP+ when an off-target sgRNA is used, since the GFP-Cas9 should still be expressed (if this is because the chain

reaction doesn't occur, this must be much more clearly explained in the text, this is not a common technique)? And likewise, why are there GFP-/RFP+ with the on-target guide.

Clarify in the S1D schematic --- there appear to be two different "b" primer sites. There is one downstream of the rightward HDR site shown as a black box, which should remain after HDR, and there is another one within the recombined DNA near the leftward HDR site. It is not clear what the a+b reaction should yield after HDR, as there are two different b primer annealing sites. Also, it would be very helpful to indicate the DNA sites on Figure S1D. This is essential for the reader to cross-compare S1D with the PCR result in 1E.

In Figure 1G, since CRISPR/Cas9 is targeting the viral RNA polymerase, this experiment should include a direct measure of effects on RNAP. If an antibody is available against the viral RNA polymerase, showing that its expression is knocked out would be helpful. If not available, the authors could perform qPCR analysis of viral transcripts, which are the direct product of the viral polymerase. Ideally, the authors should show that a cDNA rescue can be achieved (i.e. by providing RNAP refractory to CRISPR editing in trans). But at a minimum, since this experiment is being used to validate their CRISPR system, it would at least be important to show that similar results can be obtained with independent gRNAs targeting RNAP, to raise confidence that on-target effects were achieved.

Quantifying effects on viral load is a downstream effect, so while interesting, many things could disrupt this complicated process indirectly, including effects on cell fitness resulting from DNA damage. Also, the Figure axis label in 1G should be clarified - what are "relative # of particles".

An important aspect not considered in the manuscript is that CRISPR editing can have effects on the host cell fitness, in particular if there are off-target effects on the amoeba genome. But, there can even be well-defined effects on host cell fitness resulting from on-target DNA damage responses in higher organisms resulting from NHEJ, particularly when high copy # sites are targeted (for instance, PMID : 27260156). In mammalian cells, editing high copy # sites results in DNA damage and loss of host cell fitness, and as a result viral copy # achieved by lytic replication. This could confound analysis of viral copy #, since host cell death would reduce viral replication. Effects of CRISPR editing on host cell number do not appear to be controlled. What effect does CRISPR/Cas9 editing have on cell viability? Did the authors consider live cell numbers when an "on-target" gRNA is expressed vs a non-targeting control.

Figure S1C. Better define integration of "CRISPR/Cas9" in the mutagenic chain reaction strategy. Explain why the number 24 is written in the figure. Explain in the text why Cas9 integration triggers a DSB in other alleles, this is not clear. It is also critical to define for the reader of the general audience why there are 24 copies?

The manuscript should clearly state the multiplicity of infection used in all virus infection experiments. With regards to Figures 1G and S1G/H, if a high MOI is used, it should be possible that some secreted virus could have edited RNAP, despite its essentiality, because other virion within the host cell that escaped CRISPR editing could still supply RNAP in trans. Similarly, what is the eclipse time for GV replication (the time to first completion of the viral lifecycle and secretion of virus)? This should be stated, as if it is fast, this will also contribute to a situation where multiple viruses can infect the same host cell.

What is the 'screened library' in Figure 1D and mentioned in line 56. This is not well explained. Presumably this means the edited cell pool. Would clarify the term 'screened library'. Also, insertion deletion sequencing (indel seq) would be a valuable companion to the displays of several qPCR Sanger sequences. Indel seq provides quantitative readout of editing at the on-target site and is a more rigorous way to quantify the extent of on-target site editing than the way the authors currently display this result.

Figure 2. Again, please state the MOI used for the experiment. Define black/red fonts in the 2A fig legend. A schematic of the viral genome sites being targeted in Figure 2 would be very helpful. In 2C, would state in the main text or legend how clonal virus was derived. Importantly, how do sgRNAs alter host cell fitness? A nice control would be to show that a sgRNA against a particular

GV affected replication of that GF, and not another GV in the same host. That would convincingly show that effects were specific to the GV and not off-target on the host cell. Clarify in the figure 2D legend how this virus was created (what was the sgRNA that lead to this deletion?). Would clarify in the figure legend of 2E that this still refers to a comparison of wt vs delta 612-974. Clarify in 2I how essential/non-essential genes in red/blue were defined.

In Figure 3C, it is difficult to know where gene editing has been successful. The assumption is made that it is always successful, but without defining whether CRISPR has happened or not, there is potential for false negative. Similarly, large deletions, rather than precise gene KO, may complicate analyses, as multiple genes may be affected by a given perturbation.

It would be important for the authors to discuss why CRISPR targeting of the 3' end are more likely to cause deletions, based on understanding of the viral lytic cycle. This is an interesting observation, but is not sufficiently considered in the text. What could be the molecular basis for this observation. They could consider cross-comparing with PMID 31789594 or similar papers, in which CRISPR editing of a large herpesvirus genome in the lytic cycle results in large DNA deletions.

Minor Points:

In Fig 1B, would suggest labeling schematics to indicate where sgRNA targets, pam sites, etc.

Pandoravirus replication is not something that the general audience will know much about. It should be better introduced, since this is the context within which their CRISPR editing takes place. Figure 2 launches into 5' vs 3' mutations in the viral genome, but this is presented without the context of how that might relate to pandoravirus lytic replication. Additional important info should be presented to the reader to help interpret these experiments. What is the burst size – how many virion are produced per cell in the absence of editing?

Would call it PCR of the genomic locus instead of PCR on the locus.

Line 82 has two periods: double strand breaks..

Would move the map of the construct used and cartoon depicting the disruption strategy from s1c-d to the main text to help the readability.

Fig S1F. Better label the GFP and the replication compartment in the images shown. Provide an uninfected control and a no GFP-Cas9 infected cell control for comparison.

S1G. Explain why the a+c primer PCR reaction does not yield a band.

S2H. explain in the legend that Red is the WT sequence and black are PCR-amplified sequences from the experiment.

It is stated in lines 108/9 that the virus can repair breaks by NHEJ. It might be interesting to therefore investigate the effects of the DNA ligase IV inhibitor that blocks NHEJ.

I would suggest introducing headers between sections in the paper that generally correspond to the four figures.

Would relabel Figure 2G y axis. Since it's based on DNA and not proteomic analysis, would call it viral open reading frame (ORF) abundance rather than protein.

Would label genome positions on the schematic in Fig 2L that correspond to the x-axis positions listed for pneo and pmac

In Figure 2K, would be nice to clarify for the reader that the PAM is on the other complimentary strand --- CCG is listed as a PAM, but of course it's the reverse complement CGG that is the S. pneumo cas9 "NGG" PAM.

Would label the molecular wt marker sizes in Figure S2A. The expected bp sizes are listed, but the reader can't tell what the molecular weight size markers correspond to, complicating interpretation.

This statement in line 120 is difficult to follow: the assessment of the non-essential region showed an increase fitness cost upon larger deletions, as shown by a larger amount of progeny produced when the targeting site is present closer to the 3' end of the genome. Shouldn't a larger amount of progeny produced mean higher, not lower fitness?

Line 125-7, provide a reference for the expanding/contracting genome hypotheses.

The text describes cis-complementation with Figures 4F/G, but it is already a major part of 4E. This needs to be described along with 4E, before we get to 4F, since it is part of 4E. Or, the authors could move this data to follow 4F so that it is properly explained in time. Likewise, the authors should better explain in the main text how cis-complementation bypass CRISPR/Cas9 editing.

Reviewer #3 (Remarks to the Author):

This is a well-structured functional genomics study exploring the use of CRISPR/Cas9 for modifying the genomes of giant viruses. The authors have taken care to describe how Cas9 in combination with targeted gRNAs can be used to successfully edit giant viruses, citing effects on virus replication and morphogenesis. Also, they cleverly use these techniques with adequate controls to identify regions of genomic regions that are essential or dispensable for virus growth.

However, the paper lacks robust support for claims of stepwise evolution and genetic expansion of giant viruses (Fig. 4). There is a clear phenotype for mcp-deficient molivirus particles, yet the link to evolution of pandoravirus is not demonstrated within these data. The conclusions of the paper could be strengthened by examining the effect of molivirus-derived mcp introduction into pandoravirus (in relation to virus fitness and morphology).

There are also minor issues which can be addressed by the authors. Descriptions of methodology within the text can be expanded upon, for example the use of the term "selfish" gene in Fig. 1b/c is not referred to in the manuscript body. In some figures, the panel order could be simplified. Other minor amendments are included below.

Fig 1b and c - "Selfish" gene, is this referring to use of the mutagenic chain reaction strategy?

Fig 1f - The term VF should be defined.

Fig 2a - Does the red and black text signify something to the reader? A schematic of the genome showing the targeted loci to the reader could be helpful here.

Fig 2e - A log scale may be more suitable here. Can the authors comment on the magnitude of change within the manuscript text.

Fig 2i - What is the particle number relative to here?

Fig 2j top panel - It is hard to discern differences in size here.

Fig 2l - Is it possible to put numbers on the schematic in the top part of the figure? Please see comment about red and black text for Fig. 2a.

Fig 3b - n-value for Node 0 is not clear, as is the node name, a contrasting font colour is advised

Fig 3c - See comment for Fig 2i.

Fig 3e - Consider revising figure structure to label categories within the figure itself

Fig 3g - It appears that KO of one SNF2 gene still impacts replication, suggesting the genes are not entirely redundant. It would add interest if the authors commented on this within the text.

Fig 4c - Is it known what the location of endogenous MCP is during infection?

Line 186 - "we were was..." typo

Line 85-87 - "...viruses harboring such 3' large deletions were able to complete the full cycle of

replication...". Could the authors comment on whether these viruses are "piggy backing" off parental viruses? Were these viruses able to grow in pure culture, or only in instances where parent virus was also present?

Line 225 – This interpretation may need rewording. The authors clearly demonstrate the role of MCP in molivirus biogenesis, but do not show evidence of a stepwise progression from moli to pandoravirus. This should be clarified.

Fig S1a - "Partental" typo

Fig S1f – Are there more examples of GFP exclusion from the virus factories? It may be difficult for those unfamiliar with looking at virus factories to discern the structures you are referring to.

Fig S1g – What are a+b and a+c referring to? According to my understanding of Fig S1E, a+c should be detectable and not a+b? Can the authors please clarify this? For Pandoravirus, a+c is listed as larger than a+b, which differs from the other viruses listed.

Fig S2a – Ladder size markings would be helpful here

REVIEWER COMMENTS

Reviewer #1 (Remarks to the Author):

In this manuscript Bision, Legendre, co-authors and Abergel describe the use of CRISPR/Cas9 as a genomic editing tool applied to amoebal giant viruses, particularly those with nuclear phase. This study is original and certainly will inaugurate a new age on giant viruses basic studies. The application of this tool was well explored in the context of pandoravirus biology. A substantial number of constructions and experiments pointed to the main essential spots in pandoravirus genome.

Major points:

- As this work will open the windows to gene function studies on giant viruses field, it would be essential the authors provide a detailed description of materials and methods related to genomic manipulation. It would be very useful a supplemental document containing the sequences of modified GFP and its promoter, RFP, selection markers, etc. I am aware that all this information is precious and authors will apply this tool to future studies. But I see genome editing as the most consistent novelty of this work, therefore details on how to do that would be essential to grant reproducibility.

We appreciate the interest of the reviewer for the tools developed in this study and acknowledge the previous lack of details in the former version of the manuscript. In order to facilitate the delivery of the plasmids, we have now deposited the plasmids vc241, vHB8, vHB16, vHB66, vHB178 and vHB179 in addgene that will be fully available upon publication of this manuscript (accession number introduced in Table S3). Full sequences can be obtained from addgene.

- Conclusions on pandoravirus and mollivirus evolution sound very speculative. I agree that the position of some clusters of genes would suggest common evolutive features and ancestry. But even considering all results together the story sounds risky and needs more data: eg. discovery of new viruses to fulfill some gaps. The story of the manuscript don't need to rely on evolution of pandoravirus, because the use of CRISPR/Cas9 to edit giant viruses is a breakthrough to the field.

We agree with the reviewer. Considering the comments of Reviewer #1, #2 and #3, we have toned down our claims regarding the evolutionary path of these viruses and included a discussion section for better differentiation of data vs interpretation of it. We hope the revised manuscript addresses the concerns of the reviewer.

Regardless, we strongly believe that the results presented here show data that points out to a genetic expansion leading to GVs. This data resonates with the phylogeny recently accepted by the ICTV and thus, when taken together, generate a fair case regarding the evolution of giant pandoravirus from smaller icosahedral ones. Concordantly, while we have now phrased these claims in a more speculative way, we still believe it is worth mentioning and exploring in this manuscript.

- In several experiments in this work, particles counting was associated to virus fitness. Could authors explain why do they quantified total particles instead quantify infectious particles? I believe that the measure of infectious particles is specially important in the context of generation of recombination viruses. Some conclusions possibly were based considering a mix of infectious and defective particles. I see the generation of defective particles as a negative factor for viral fitness.

We agree with the reviewer. We have originally used the burst size of the different mutants as an estimate of fitness since the large size of pandoravirus particles allows an easy quantification. Moreover, pandoravirus presents low infection rates and only approx. 1% of the particles are infectious. This characteristic of pandoravirus leaves a small window to assess differences between wild-type and mutant viruses. Regardless, we have now calculated and included data regarding the infective dose 50 (ID50) of all single knockout generated in this study (included in figure S4c).

Minor points:

- abstract: is the "fourth domain of life" worth of note? This theory has been refuted by several comprehensive evolutionary studies.

The first sentence of the abstract has now been modified as follows:

"Giant viruses (GVs) are a hotspot of unresolved controversies since their discovery, including the definition of "Virus" and their origin."

- line 41: metabolism is a generic term. Please, consider specify.

Corrected. Now specified as energy metabolism.

- lines 43-44: "blurred the division between cellular living organisms and viruses". This sentence sounds exaggerated. The authors group published papers on the discovery of some of the most awesome giant viruses ever... and those entities were described as viruses since the beginning by them. I mean, with good sequencing and cycle imaging it is possible to differentiate a giant virus from a cellular organism: FtsZ genes, ribosomes r and proteins, capsid and hallmark genes, eclipse phase, etc, analyzed together in a context.

We agree with the reviewer. We have now eliminated such sentence from the manuscript.

- line 47: is it necessary to get some federal/local authorization to manipulate the genome of a given organism in France? In some countries this is necessary. If yes, please consider to provide the authorization number.

Yes and we did receive an Genetically Modified Organism agreement from the Ministry of Higher Education, Research and Innovation. Agreement DUO: 5975

- Fig 1C: please provide more details on how the fluorescence was measured.

Corrected. We have now expanded the materials and methods related to Fluorescence imaging.

- line 64: Medusavirus does not encode to a RNA pol.

Corrected. We thank the reviewer for spotting this error.

- Fig S1F: why black and white figures?

Now Fig S2b. It has now been modified to green.

Reviewer #2 (Remarks to the Author):

Bisio and co-authors presents new methods to use CRISPR/Cas9 to edit giant pandoravirus genomes and study roles of targeted regions in virus replication. While there has been considerable interest in the lifecycle of these megabase genome size viruses, genetic tools have been lacking to enable their manipulation by CRISPR/Cas9 technology. The authors introduce *S. pyogenes* Cas9 together with pU6-driven guide RNAs (gRNA). They show that they are able to edit a model RFP gene in amoeba by CRISPR, and that they can employ a chain reaction approach to increase homologous recombination (though this is not well described). They then test effect on GV that replicate in the nucleus vs cytoplasm, and find that the former are targetable by CRISPR/Cas9. CRISPR approaches are then used to target viral genes, to interrogate their essentiality in viral replication. Effects on viral replication were stronger for gRNA targeting the 5' end of the genome and inducing NHEJ repair, whereas targeting of the gene poor 3' viral genome had comparatively little effects on fitness, despite inducing in some cases large deletions. This interesting difference is not well described at the molecular level – why would targeting of the 3' region result in large deletions, based on understanding of the viral lytic replication mechanisms? The authors then consider pandoravirus genome organization, gene density, and effects of targeting different genomic regions/genes located in the essential core of the genome. Finally, CRISPR is used to test effects of MCP KO on tegument biosynthesis and virus replication.

Overall, the manuscript represents an interesting application of CRISPR/Cas9 technology to studies of GV replication. The authors present strong evidence to support the ability of CRISPR/Cas9 to edit GV genomes by HDR or NHEJ. However, there are a number of issues that would need to be addressed to increase robustness and readability of the manuscript, particularly for a general audience.

Major points:

There is jargon that is not defined and would be best avoided. This makes the manuscript difficult to read. A good example is the term Selfish gene in Fig 1C, which is never defined in the main text or legend. What does this mean? Would take out this term and instead specifically define what is being tested. As another important example, the term mutagenic chain reaction is used in line 57, with no explanation of what this means. This is confusing. This should be much more clearly explained in the text and legend, and not left to reference 15. Likewise, the terms off target and on target are used frequently. Instead of using these potentially confusing terms, define in the text/legend what the “off-target” sgRNAs are actually targeting. You want to use a good control guide that has on-target effects on a control site. And, an on-target sgRNA can have off-target effects elsewhere. I would instead re-name the on-target sgRNA as anti-RFP or whatever it is designed to target, and the off-target sgRNA to indicate its on-target site that allows it to serve as a control.

We acknowledge the use of confusing and redundant terminology in the previous version of the manuscript. We have now scrutinized the manuscript in order to identify any unnecessary jargon. We hope the manuscript would be easier to follow for the general audience in its current state. We have also followed the suggestion of the reviewer and specified the targeting sequence of off-target gRNAs in the figure legends.

Since RFP is heavily used for foundational CRISPR experiments in Figure 1, it should be clearly defined in the main text how RFP is expressed in the amoeba. It is said to be episomal, but how many copies is the episome? What is the episome that is used? Why did the authors chose to target an episomal gene and not an amoeba gene for which they have an antibody?

Acanthamoeba castellanii is a highly complex organism regarding its genetic content. First, *A. castellanii* is highly polyploid, with a n number of approximately 25¹. Moreover, these organisms are amitotic and segregate chromosomes randomly^{2,3}, leading to high proportions of aneuploidy. Considering this complexity, we decided to set up the system using a vector expressing mRFP (vc241, the sequence of vc241 has now been deposited in addgene) for simplification of the readout. Regardless, the copy number of the vector in each individual amoeba is likely different due to the random segregation of DNA.

Overall, we agree with the reviewer that the targeting of an amoeba gene would be essential to demonstrate the feasibility of genome modification by CRISPR/Cas9. Unfortunately, we do not possess any antibody raised against a non-essential gene in *A. castellanii*. Moreover, the high polyploidy precludes endogenous tagging since modification of all alleles is unlikely. Coherently, we have opted for generating a knockout of a gene with known phenotypic consequences upon downregulation. We have now included in the new version of the manuscript the targeting of the cellulose synthase of *A. castellanii* (Figure 1g-j and Figure S1f-i).

This example allows to demonstrate the high efficiency of CRISPR/Cas9 to modify the genome of *A. castellanii*. *A. castellanii* encodes 3 genes of the cellulose synthase, each of them encoded in approximately 25 copies¹ (due to the polyploidy of these organisms). In sum, since 2 of the genes were targeted by 2 different gRNA and the last gene by one gRNA, this renders 125 modified loci.

We hope the potential of CRISPR/Cas9 to modify the genome of these organisms is more clearly demonstrated with this new data. Regardless, we are fully aware that the system will need further improvements in order to reach quality levels similar to the one achieved in model organisms. We have also clearly stated this in the manuscript. We hope the joint effort of the *Acanthamoeba* field will capitalize on our results to improve the manipulation of this organism.

Figure 1C is hard to follow and should be better explained. Why are 100% of cells RFP+ but GFP- in column 1? Does this mean they were not transfected? Then, in column 3, why are some cells GFP- and RFP-? This is confusing. If they are RFP-, presumably the CRISPR worked. But why then did they lose the GFP-Cas9 signal? Likewise, why aren't all of the cells GFP+/RFP+ when an off-target sgRNA is used, since the GFP-Cas9 should still be expressed (if this is because the chain reaction doesn't occur, this must be much more clearly explained in the text, this is not a common technique)? And likewise, why are there GFP-/RFP+ with the on-target guide.

We apologize for the lack of information in the previous version of the manuscript.

First, the first column shows the quantification of the parental fluorescent cells (they do not express GFP-Cas9-NLS). We have now indicated the presence or absence of GFP-Cas9-NLS in the figure.

Second, the presence of double-negative cells is likely explained by the amitotic nuclear division of these organisms. The following sentence has been introduced in the manuscript:

“Importantly, the presence of double-negative amoebas is likely explained by the amitotic nuclear division of these organisms, leading to aneuploidy and the potential loss of genetic material”

Third, the presence of GFP-/RFP+ cells in both on-target or off-target would indicate the acquisition of the selection cassette present in plasmid encoding for GFP-Cas9-NLS but no expression of GFP-cas9-NLS. We have not followed these cells in order to assess the genotype associated to this phenomenon. Recombination events, mutagenesis that impede the expression of GFP-Cas9 or abnormal expression of the gene are possible explanations.

Clarify in the S1D schematic --- there appear to be two different “b” primer sites. There is one downstream of the rightward HDR site shown as a black box, which should remain after HDR, and there is another one within the recombined DNA near the leftward HDR site. It is not clear what the a+b reaction should yield after HDR, as there are two different b primer annealing sites. Also, it would be very helpful to indicate the DNA sites on Figure S1D. This is essential for the reader to cross-compare S1D with the PCR result in 1E. Figure S1D is now Figure 1e.

Indeed, the primer b can anneal both in the unmodified locus and the recombinant one. We have better clarified this aspect. We hope the changes help a better understanding of the genotyping procedure. PCR product size have been specified in the figure legend in the previous version of the manuscript.

In Figure 1G, since CRISPR/Cas9 is targeting the viral RNA polymerase, this experiment should include a direct measure of effects on RNAP. If an antibody is available against the viral RNA polymerase, showing that its expression is knocked out would be helpful. If not available, the authors could perform qPCR analysis of viral transcripts, which are the direct product of the viral polymerase. Ideally, the authors should show that a cDNA rescue can be achieved (i.e. by providing RNAP refractory to CRISPR editing in trans). But at a minimum, since this experiment is being used to validate their CRISPR system, it would at least be important to show that similar results can be obtained with independent gRNAs targeting RNAP, to raise confidence that on-target effects were achieved.

Figure 1G is now Figure 2b.

We agree with the reviewer. It is important to stand out that the scope of this manuscript is not to study in detail the phenotypic consequences of the RNAP from nuclear giant viruses but rather expand into the methodology that would allow this kind of studies.

Regardless, in order to address this point, we have generated a vector to endogenously tag genes in pandoravirus. These new vectors have now been deposited to addgene. Importantly, we have endogenously tagged *rbp2*, which was previously successfully c-terminally tagged in vaccinia virus⁴. We demonstrated that endogenously tagged *rbp2* is expressed and that the presence of on-target gRNA led to the disappearance of the protein. This data is now included in Figure 2d-g and Figure S2d.

Moreover, we have also included two other pairs of gRNA targeting *rbp1* in order to rule out potential off-target effects. This data is now included in Figure 2c.

We hope these additional experiments increase the strength of the developed methodology.

Quantifying effects on viral load is a downstream effect, so while interesting, many things could disrupt this complicated process indirectly, including effects on cell fitness resulting from DNA damage. Also, the Figure axis label in 1G should be clarified – what are “relative # of particles”. Figure 1G is now Figure 2b.

We agree with the reviewer. Regardless, the major point of the current manuscript is not to assess the role of any specific gene (except for the major capsid protein of mollivirus) but rather the development of genetic tools for tractability of GVs. We believe that the burst size is a good estimate of such “fitness score” and serves for the purpose of this manuscript. The detailed characterization of each of the mutant falls out of the scope of this manuscript. Regardless, following the suggestion of reviewer #1, we have now also included the ID50 of each of the single knock out generated in this study.

Please see following responses for clarification of potential side effects of CRISPR/Cas9. Further clarification of the meaning of the axis label has also been provided in the figure legend.

An important aspect not considered in the manuscript is that CRISPR editing can have effects on the host cell fitness, in particular if there are off-target effects on the amoeba genome. But, there can even be well-defined effects on host cell fitness resulting from on-target DNA damage responses in higher organisms resulting from NHEJ, particularly when high copy # sites are targeted (for instance, PMID : 27260156). In mammalian cells, editing high copy # sites results in DNA damage and loss of host cell fitness, and as a result viral copy # achieved by lytic replication. This could confound analysis of viral copy #, since host cell death would reduce viral replication. Effects of CRISPR editing on host cell number do not appear to be controlled. What effect does CRISPR/Cas9 editing have on cell viability? Did the authors consider live cell numbers when an “on-target” gRNA is expressed vs a non-targeting control.

We agree with the reviewer. We have now included measures of doubling time of the amoebas expressing Cas9 (Figure S1c) and % of dead cells (Figure S1d). No effects were observed in any situation.

We have also included the generation of a knock out in the *cellulose synthase* genes of *A. castellanii* which, besides expressing gRNAs on target and Cas9 stably, do not possess any phenotype on cellular growth. It is also worth mentioning that Cas9 and the gRNAs are stably expressed in the amoeba for weeks before any experiment is performed, including viral infections. This time includes the transfection and selection of the amoebas with the desired plasmid. Double strand breaks and repair likely occurs during this period of time, which generate ultimately modifications on the gRNA targets site.

The second concern of the reviewer has previously been assessed in the manuscript. In particular, we engineered viruses for the cis-complementation of MCP harboring 2 copies of the gene, one resistant to Cas9 and the second one susceptible to it. This experiment was designed this way in order to address the effect of the double strand break in the fitness of the virus (directly or indirectly through host signaling associated to DNA damage). Since the cis-complemented viruses were able to grow and restored over 70% of the burst size of the viruses (even in the presence of a double-strand break caused by Cas9), we can confidently ensure that phenotypes observed are largely due to the gene targeted rather than the double-strand break generated in the genome of the virus.

Figure S1C. Better define integration of “CRISPR/Cas9” in the mutagenic chain reaction strategy. Explain why the number 24 is written in the figure. Explain in the text why Cas9 integration triggers a DSB in other alleles, this is not clear. It is also critical to define for the reader of the general audience why there are 24 copies?

Figure S1C is now Figure S1e.

We apologize for the lack of information. We have now expanded the explanation in the figure legend of Figure S1e. The number 24 refers to the polyploidy of *A. castellanii* (25n approx.). We have now clarified this in the manuscript.

The manuscript should clearly state the multiplicity of infection used in all virus infection experiments. With regards to Figures 1G and S1G/H, if a high MOI is used, it should be possible that some secreted virus could have edited RNAP, despite its essentiality, because other virion within the host cell that escaped CRISPR editing could still supply RNAP in trans. Similarly, what is the eclipse time for GV replication (the time to first completion of the viral lifecycle and secretion of virus)? This should be stated, as if it is fast, this will also contribute to a situation where multiple viruses can infected the same host cell. Figure 1G is now Figure 2b. Figure S1G/H are now Figure S2c.

We agree with the reviewer. This is the reason why we utilize a MOI of 1 for most of the experiments described in this manuscript. Regardless, it is still possible that there would be some low level of transcomplementation between viruses but we do not expect this to affect significantly the results obtained from the burst size.

We have previously included the information of the MOI in the methods section. Yet, we have now included this information on the figure legends to facilitate access and clarity.

Finally, we have included information regarding the replication cycle of pandoravirus in the introduction of the manuscript as suggested by this reviewer.

What is the ‘screened library’ in Figure 1D and mentioned in line 56. This is not well explained. Presumably this means the edited cell pool. Would clarify the term ‘screened library’. Also, insertion deletion sequencing (indel seq) would be a valuable companion to the displays of several qPCR Sanger sequences. Indel seq provides quantitative readout of editing at the on-target site and is a more rigorous way to quantify the extent of on-target site editing than the way the authors currently display this result.

In order to genotype the modified locus of the mutants generated in this study, the locus targeted by the gRNA were amplified by PCR, cloned into a TA vector (pGEM-T easy) and 10 single clones were isolated and send for sequencing. We have now clearly stated this information in the manuscript. While we agree that indel seq would be a more precise way to assess the genotype of the mutants, we believe that the screening of a library gives a reliable estimate which serves its purpose for this manuscript.

Figure 2. Again, please state the MOI used for the experiment. Define black/red fonts in the 2A fig legend. A schematic of the viral genome sites being targeted in Figure 2 would be very helpful. In 2C, would state in the main text or legend how clonal virus was derived.

Figure 2A is now Figure 3a. Figure 2c is now Figure 3c.

We have previously included the information of the MOI and cloning strategy in the methods section. Yet, we have now included this information on the figure legends to facilitate access and clarity.

Importantly, how do sgRNAs alter host cell fitness? A nice control would be to show that a sgRNA against a particular GV affected replication of that GF, and not another GV in the same host. That would convincingly show that effects were specific to the GV and not off-target on the host cell.

We agree with the reviewer with the proposed experiment. We have now included the information of the identity of the off-target guides. The off-target guide used in most of the experiments for pandoravirus are the gRNAs used to target the *rbp1/mcp* of mollivirus.

Clarify in the figure 2D legend how this virus was created (what was the sgRNA that led to this deletion?). Would clarify in the figure legend of 2E that this still refers to a comparison of wt vs delta 612-974. Clarify in 2I how essential/non-essential genes in red/blue were defined.

Figure 2D is now Figure 3d. Figure 2E is now Figure 3e.

All corrections have also been incorporated.

In Figure 3C, it is difficult to know where gene editing has been successful. The assumption is made that it is always successful, but without defining whether CRISPR has happened or not, there is potential for false negative. Similarly, large deletions, rather than precise gene KO, may complicate analyses, as multiple genes may be affected by a given perturbation.

Figure 3C is now Figure 4c.

We have now modified the PCR showed in figure S4 to demonstrate deletions of the expected size between gRNA targeting sites. This data clearly demonstrates discrete modifications of the genome rather than large deletions.

It would be important for the authors to discuss why CRISPR targeting of the 3' end are more likely to cause deletions, based on understanding of the viral lytic cycle. This is an interesting observation, but is not sufficiently considered in the text. What could be the molecular basis for this observation. They could consider cross-comparing with PMID 31789594 or similar papers, in which CRISPR editing of a large herpesvirus genome in the lytic cycle results in large DNA deletions.

We have now extended the discussion regarding the possible explanations for the differences in deletion sizes at different locations of the genome. We hope the new manuscript allows a better interpretation of the data.

Minor Points:

In Fig 1B, would suggest labeling schematics to indicate where sgRNA targets, pam sites, etc.

Pandoravirus replication is not something that the general audience will know much about. It should be better introduced, since this is the context within which their CRISPR editing takes place. Figure 2 launches into 5' vs 3' mutations in the viral genome, but this is presented without the context of how that might relate to pandoravirus lytic replication. Additional important info should be presented to the reader to help interpret these experiments. What is the burst size – how many virion are produced per cell in the absence of editing?

Figure 2 is now Figure 3.

We have now expanded the introduction to include information regarding the replication cycle of pandoravirus.

Would call it PCR of the genomic locus instead of PCR on the locus.

Fixed.

Line 82 has two periods: double strand breaks..

Fixed.

Would move the map of the construct used and cartoon depicting the disruption strategy from s1c-d to the main text to help the readability.

Done. New Figure 1e.

Fig S1F. Better label the GFP and the replication compartment in the images shown. Provide an uninfected control and a no GFP-Cas9 infected cell control for comparison.

Figure S1F is now Figure S2b.

Done.

S1G. Explain why the a+c primer PCR reaction does not yield a band.

Figure S1G is now Figure S2c.

Done.

S2H. explain in the legend that Red is the WT sequence and black are PCR-amplified sequences from the experiment.

Figure S2H is now Figure S2c.

Done.

It is stated in lines 108/9 that the virus can repair breaks by NHEJ. It might be interesting to therefore investigate the effects of the DNA ligase IV inhibitor that blocks NHEJ.

We agree with the reviewer and we are grateful for the suggestion. We will include studies with these inhibitors in follow-up studies, which might also facilitate homologous recombination to prompt the mutagenesis chain reaction for modifications of *A. castellanii* genome.

I would suggest introducing headers between sections in the paper that generally correspond to the four figures.

Done.

Would relabel Figure 2G y axis. Since it's based on DNA and not proteomic analysis, would call it viral open reading frame (ORF) abundance rather than protein.

Figure 2G is now Figure 3g.

Done.

Would label genome positions on the schematic in Fig 2L that correspond to the x-axis positions listed for pneo and pmac

Figure 2L is now Figure 3I.

Done.

In Figure 2K, would be nice to clarify for the reader that the PAM is on the other complimentary strand --- CCG is listed as a PAM, but of course it's the reverse complement CGG that is the *S. pneumo* cas9 "NGG" PAM.

Figure 2K is now Figure 3k.

Done.

Would label the molecular wt marker sizes in Figure S2A. The expected bp sizes are listed, but the reader can't tell what the molecular weight size markers correspond to, complicating interpretation.

Figure S2A is now Figure S3a.

Fixed.

This statement in line 120 is difficult to follow: the assessment of the non-essential region showed an increase fitness cost upon larger deletions, as shown by a larger amount of progeny produced when the targeting site is present closer to the 3' end of the genome. Shouldn't a larger amount of progeny produced mean higher, not lower fitness?

We agree with the reviewer that such phrase was unnecessarily deconvoluted and it has now been removed from the manuscript.

Line 125-7, provide a reference for the expanding/contracting genome hypotheses.

Fixed.

The text describes cis-complementation with Figures 4F/G, but it is already a major part of 4E. This needs to be described along with 4E, before we get to 4F, since it is part of 4E. Or, the authors could move this data to follow 4F so that it is properly explained in time. Likewise, the authors should better explain in the main text how cis-complementation bypass CRISPRR/Cas9 editing.

Figure 4E/F/G are now Figure 5e-g

We have now included the following sentence in the figure legend of figure 5e:

“Refer to figure Fig. 5f for details on the strategy for cis-complementation.”

Reviewer #3 (Remarks to the Author):

This is a well-structured functional genomics study exploring the use of CRISPR/Cas9 for modifying the genomes of giant viruses. The authors have taken care to describe how Cas9 in combination with targeted gRNAs can be used to successfully edit giant viruses, citing effects on virus replication and morphogenesis. Also, they cleverly use these techniques with adequate controls to identify regions of genomic regions that are essential or dispensable for virus growth.

However, the paper lacks robust support for claims of stepwise evolution and genetic expansion of giant viruses (Fig. 4). There is a clear phenotype for mcp-deficient molivirus particles, yet the link to evolution of pandoravirus is not demonstrated within these data. The conclusions of the paper could be strengthened by examining the effect of molivirus-derived mcp introduction into pandoravirus (in relation to virus fitness and morphology).

We agree with this reviewer.

We have previously performed the experiment proposed by this reviewer. As a result, pandoravirus is not capable of incorporating the RFP-MCP or MCP-RFP of mollivirus into their virions (not shown).

Regardless, we have now modified the manuscript to assess such concerns. Particularly, we have included the following sentence in the manuscript:

“If pandoravirus ancestors possessed mollivirus-like particles and which molecular changes allowed the dispensability of the scaffolding functions of MCP are yet to be studied.”

Our future efforts will be directed to dissect molecular interaction at the tegument of mollivirus and pandoravirus in order to grasp a better understanding of the evolution of such particles.

There are also minor issues which can be addressed by the authors. Descriptions of methodology within the text can be expanded upon, for example the use of the term “selfish” gene in Fig. 1b/c is not referred to in the manuscript body. In some figures, the panel order could be simplified. Other minor amendments are included below.

Fig 1b and c - “Selfish” gene, is this referring to use of the mutagenic chain reaction strategy?

Corrected. We apologize for the mixed jargon.

Fig 1f – The term VF should be defined.

Fig 1f is now Fig 2a.

Corrected.

Fig 2a – Does the red and black text signify something to the reader? A schematic of the genome showing the targeted loci to the reader could be helpful here.

Fig 2a is now Fig 3a.

Fixed. A schematic representation has been also included.

Fig 2e – A log scale may be more suitable here. Can the authors comment on the magnitude of change within the manuscript text.

Fig 2e is now Fig 3e.

We have considered a log scale and we believe it makes difficult the interpretation of the data. We have now added the following comment on the result section:

“Fitness reduction of pandoravirus was highly dependent on the targeted gene locus (Fig. 3a) and ranged from 100-times in particle numbers when genes at the 5’ of the genome were targeted, to no fitness cost when genes at the 3’ of the genome were targeted.”

Fig 2i – What is the particle number relative to here?

Fig 2i is now Fig 3i.

Fixed. We apologize for the lack of information.

Fig 2j top panel – It is hard to discern differences in size here.

Fig 2j is now Fig 3j.

A slight shift is present but due to the fact that the difference in size is only of 50bp while the PCR product size is about 1200 it is indeed hard to discern. We trust that the results of sequencing are sufficient to convince the reviewer of the presence of the desired deletion.

Fig 2l – Is it possible to put numbers on the schematic in the top part of the figure? Please see comment about red and black text for Fig. 2a.

Fig 2l is now Fig 3l.

Done. Essential genes in *p. neocaledonia* are highlighted in red.

Fig 3b – n-value for Node 0 is not clear, as is the node name, a contrasting font colour is advised

Fig 3b is now Fig 4b.

Fixed.

Fig 3c – See comment for Fig 2i.

Fig 3c is now Fig 4c.

Fixed.

Fig 3e – Consider revising figure structure to label categories within the figure itself

Fig 3e is now Fig4e.

Done.

Fig 3g – It appears that KO of one SNF2 gene still impacts replication, suggesting the genes are not entirely redundant. It would add interest if the authors commented on this within the text.

Fig 3g is now Fig 4g.

We have now included the following sentence in the manuscript:

“Regardless, both genes are not strictly redundant since a fitness cost associated with single deletion was observed.”

Fig 4c – Is it known what the location of endogenous MCP is during infection?

Fig 4c is now Fig 5c.

We have not further studied the intracellular location of the MCP. Future effort will be done to perform correlative electron microscopy and assess its localization in greater details.

Line 186 – “we were was...” typo

Corrected.

Line 85-87 – “...viruses harboring such 3’ large deletions were able to complete the full cycle of replication...”. Could the authors comment on whether these viruses are “piggy backing” off parental viruses? Were these viruses able to grow in pure culture, or only in instances where parent virus was also present?

Such viruses were cloned. Concordantly, no wild type viruses are present in the culture. We are currently growing these viruses in continuous culture in order to assess if fitness would change across generations. This corresponds to follow-up studies to this work.

Line 225 – This interpretation may need rewording. The authors clearly demonstrate the role of MCP in molivirus biogenesis, but do not show evidence of a stepwise progression from moli to pandoravirus. This should be clarified.

We thank the reviewer for rising this note of caution. We have now reworded the conclusion as followed:

“If pandoravirus ancestors possessed mollivirus-like particles and which molecular changes allowed the dispensability of the scaffolding functions of MCP are yet to be studied.”

Fig S1a - "Parental" typo

Fixed.

Fig S1f – Are there more examples of GFP exclusion from the virus factories? It may be difficult for those unfamiliar with looking at virus factories to discern the structures you are referring to.

Fig S1f is now Fig S2b.

To our knowledge, most bacteriophages avoid destruction mediated by CRISPR encoding anti-CRISPR proteins that inhibit CRISPR/Cas systems. We are only aware of jumbo phage like Φ KZ utilizing this strategy for protection of DNA targeting⁵.

We have now indicated the location of the viral factory by an arrow.

Fig S1g – What are a+b and a+c referring to? According to my understanding of Fig S1E, a+c should be detectable and not a+b? Can the authors please clarify this? For Pandoravirus, a+c is listed as larger than a+b, which differs from the other viruses listed.

Fig S1g is now Fig S2c.

We apologize for the lack of information. The PCR amplicon size previously shown for the primers a+c corresponds to the size upon deletion. We have now corrected this problem and explained why there is no product in these lines.

Fig S2a – Ladder size markings would be helpful here

Fig S2a is now Fig S3a.

Added.

- 1 Byers, T. J. Molecular biology of DNA in Acanthamoeba, Amoeba, Entamoeba, and Naegleria. *Int Rev Cytol* **99**, 311-341, doi:10.1016/s0074-7696(08)61430-8 (1986).
- 2 Cervantes, M. D., Coyne, R. S., Xi, X. & Yao, M. C. The condensin complex is essential for amitotic segregation of bulk chromosomes, but not nucleoli, in the ciliate *Tetrahymena thermophila*. *Mol Cell Biol* **26**, 4690-4700, doi:10.1128/MCB.02315-05 (2006).
- 3 Gicquaud, C. & Tremblay, N. Observations with Hoechst Staining of Amitosis in *Acanthamoeba Castellanii*. *J Protozool* **38**, 221-224, doi:DOI 10.1111/j.1550-7408.1991.tb04432.x (1991).
- 4 Grimm, C. *et al.* Structural Basis of Poxvirus Transcription: Vaccinia RNA Polymerase Complexes. *Cell* **179**, 1537-1550 e1519, doi:10.1016/j.cell.2019.11.024 (2019).
- 5 Mendoza, S. D. *et al.* A bacteriophage nucleus-like compartment shields DNA from CRISPR nucleases. *Nature* **577**, 244-248, doi:10.1038/s41586-019-1786-y (2020).

Reviewer comments, second round –

Reviewer #1 (Remarks to the Author):

The authors addressed all points, including improvements on methodological description. As I had mentioned in the first round of review, this work will open the windows for future studies on giant viruses gene function. It deserves to be accepted.

Reviewer #2 (Remarks to the Author):

The revised manuscript is considerably improved. I read the revision carefully and they did a very nice job of responding and revising the manuscript

Reviewer #3 (Remarks to the Author):

Bisio et al. have produced a well-structured study exploring the use of CRISPR/Cas9 for modifying the genomes of giant viruses. They have satisfactorily addressed all of the previously raised concerns. Significant improvements have been made to the information provided in the introduction and wording describing the previous Fig. 4. General clarity has been improved upon for readers.

REVIEWERS' COMMENTS

Reviewer #1 (Remarks to the Author):

The authors addressed all points, including improvements on methodological description. As I had mentioned in the first round of review, this work will open the windows for future studies on giant viruses gene function. It deserves to be accepted.

We thank the reviewer for the comments and interest in the manuscript.

Reviewer #2 (Remarks to the Author):

The revised manuscript is considerably improved. I read the revision carefully and they did a very nice job of responding and revising the manuscript

We thank the reviewer for the comments and interest in the manuscript.

Reviewer #3 (Remarks to the Author):

Bisio et al. have produced a well-structured study exploring the use of CRISPR/Cas9 for modifying the genomes of giant viruses. They have satisfactorily addressed all of the previously raised concerns. Significant improvements have been made to the information provided in the introduction and wording describing the previous Fig. 4. General clarity has been improved upon for readers.

We thank the reviewer for the comments and interest in the manuscript.